# A Kernelised Stein Statistic for Assessing Implicit Generative Models

**Wenkai Xu**
Department of Statistics
University of Oxford
wenkai.xu@stats.ox.ac.uk

**Gesine Reinert**
Department of Statistics
University of Oxford
and Alan Turing Institute
reinert@stats.ox.ac.uk

## Abstract

Synthetic data generation has become a key ingredient for training machine learning procedures, addressing tasks such as data augmentation, analysing privacy-sensitive data, or visualising representative samples. Assessing the quality of such synthetic data generators hence has to be addressed. As (deep) generative models for synthetic data often do not admit explicit probability distributions, classical statistical procedures for assessing model goodness-of-fit may not be applicable. In this paper, we propose a principled procedure to assess the quality of a synthetic data generator. The procedure is a kernelised Stein discrepancy (KSD)-type test which is based on a non-parametric Stein operator for the synthetic data generator of interest. This operator is estimated from samples which are obtained from the synthetic data generator and hence can be applied even when the model is only implicit. In contrast to classical testing, the sample size from the synthetic data generator can be as large as desired, while the size of the observed data which the generator aims to emulate is fixed. Experimental results on synthetic distributions and trained generative models on synthetic and real datasets illustrate that the method shows improved power performance compared to existing approaches.

## 1   Introduction

Synthetic data capturing main features of the original dataset are of particular interest for machine learning methods. The use of original dataset for machine learning tasks can be problematic or even prohibitive in certain scenarios, e.g. under authority regularisation on privacy-sensitive information, training models on small-sample dataset, or calibrating models with imbalanced groups. High quality synthetic data generation procedures surpass some of these challenges by creating de-identified data to preserve privacy and to augment small or imbalanced datasets. Training deep generative models has been widely studied in recent years [Kingma and Welling, 2013, Radford et al., 2015, Song and Kingma, 2021] and methods such as those based on Generative Adversarial Networks (GANs) [Goodfellow et al., 2014] provide powerful approaches that learn to generate synthetic data which resemble the original data distributions. However, these deep generative models usually do not provide theoretical guarantees on the *goodness-of-fit* to the original data [Creswell et al., 2018].

To the best of our knowledge, existing mainstream developments for deep generative models [Song and Ermon, 2020, Li et al., 2017] do not provide a systematic approach to assess the quality of the synthetic samples. Instead, heuristic methods are applied; e.g. for image data, the quality of samples are generally decided via visual comparisons. The study of the training quality has been relying largely on the specific choice of training loss, which does not directly translate into a measure of sample quality; in the case of the log-likelihood see for example Theis et al. [2015]. Common quality assessment measures for implicit generative models, on images for example, include Inception Scores

(IS) [Salimans et al., 2016] and Fréchet Inception Distance (FID) [Heusel et al., 2017], which are motivated by human inception systems in the visual cortex and pooling [Wang et al., 2004]. Bińkowski et al. [2018] pointed out issues for IS and FID and developed the Kernel Inception Distance (KID) for more general datasets. Although these scores can be used for for comparisons, they do not provide a statistical significance test which would assess whether a deemed *good* generative model is "*good enough*". A key stumbling block is that the distribution from which a synthetic method generates samples is not available; one only ever observes samples from it.

For models in which the density is known explicitly, at least up to a normalising constant, some assessment methods are available. Gorham and Mackey [2015] proposed to assess sample quality using discrepancy measures based on Stein's method; Chwialkowski et al. [2016] and Liu et al. [2016] created a kernelised test statistic called *kernelised Stein discrepancy* (KSD) based on Stein operators and used the unit ball of an RKHS as test functions. Schrab et al. [2022] assesses the quality of generative models on the MNIST image dataset from LeCun et al. [1995] using an aggregated kernel Stein discrepancy (KSDAgg) test; still an explicit density is required. The only available implicit goodness-of-fit test, AgraSSt [Xu and Reinert, 2022], applies only to generators of finite graphs; it is also of KSD form and makes extensive use of the discrete and finite nature of the problem. To date, quality assessment procedures of *implicit* deep generative models for continuous data remains unresolved. This paper addresses this gap.

The underlying idea can be sketched as follows. Traditionally, given a set of $n$ observations, each in $\mathbb{R}^m$, one would estimate the distribution of these observations from the data and then check whether the synthetic data can be viewed as coming from the data distribution. Here instead we characterise the distribution which is generated possibly implicitly from the synthetic data generator, and then test whether the observed data can be viewed as coming from the synthetic data distribution. The advantage of this approach is that while the observed sample size $n$ may be fairly small, the synthetic data distribution can be estimated to any desirable level of accuracy by generating a large number of samples from it. Similarly to the works mentioned in the previous paragraph for goodness-of-fit tests, we use a KSD approach, based on a Stein operator which characterises the synthetic data distribution. As the synthetic data generator is usually implicit, this Stein operator is not available. We show however that under mild assumptions it can be estimated from synthetic data samples to any desired level of accuracy.

**Our contributions**   We introduce a method to assess (deep) generative models, which are often *black-box* approaches, when the underlying probability distribution is continuous, usually in high-dimensions. To this purpose, we develop a non-parametric Stein operator and the corresponding non-parametric kernel Stein discrepancies (NP-KSD), based on estimating conditional score functions. Moreover, we give theoretical guarantees for NP-KSD.

This paper is structured as follows. We start with a review of Stein's method and KSD goodness-of-fit tests for explicit models in Section 2 before in Section 3 we introduce NP-KSD and analyse the model assessment procedures. We show results of experiments in Section 4 and conclude with future directions in Section 5. Theoretical underpinnings, and additional results are provided in the supplementary material. The code is available at `https://github.com/wenkaixl/npksd.git`.

## 2   Stein's method and kernel Stein discrepancy tests

**Stein identities, equations, and operators**   Stein's method [Stein, 1972] provides an elegant tool to characterise distributions via *Stein operators*, which can be used to assess distances between probability distributions, see for example Barbour and Chen [2005], Mijoule et al. [2021]. Given a distribution $q$, an operator $\mathcal{A}_q$ is called a Stein operator w.r.t. $q$ and *Stein class* $\mathcal{F}$ if the following Stein identity holds for any *test function* $f \in \mathcal{F}$: $\mathbb{E}_q[\mathcal{A}_q f] = 0$. For a test function $h$ one then aims to find a function $f = f_h \in \mathcal{F}$ which solves the *Stein equation*

$$\mathcal{A}_q f(\boldsymbol{x}) = h(\boldsymbol{x}) - \mathbb{E}_q[h(\boldsymbol{x})]. \tag{1}$$

Then for any distribution $p$, taking expectations $\mathbb{E}_p$ in Eq. 1 assesses the distance $|\mathbb{E}_p h - \mathbb{E}_q h|$ through $|\mathbb{E}_p \mathcal{A}_q f|$, an expression in which randomness enters only through the distribution $p$.

When the density function $q$ is given explicitly, with smooth support $\Omega_q \subset \mathbb{R}^m$, is differentiable and vanishes at the boundary of $\Omega_q$, a common choice of Stein operator in the literature is based on the

score-function $\mathbf{s}_q = \nabla \log q$ (with the convention that $\mathbf{s}_q \equiv 0$ outside of $\Omega_q$), see for example Mijoule et al. [2021]. Here, $\nabla$ denotes the gradient operator and taken to be a column vector. The *score-Stein operator*[1] is the vector-valued operator acting on (vector-valued) function $\mathbf{f}$,

$$\mathcal{A}_q \mathbf{f}(\boldsymbol{x}) = \mathbf{f}(\boldsymbol{x})^\top \nabla \log q(\boldsymbol{x}) + \nabla \cdot \mathbf{f}(\boldsymbol{x}), \tag{2}$$

and the Stein identity $\mathbb{E}_q[\mathcal{A}_q f] = 0$ holds for functions $f$ which belong to the so-called *canonical Stein class* defined in Mijoule et al. [2021], Definition 3.2. As it requires knowledge of the density $q$ only via its score function, this Stein operator is particularly useful for unnormalised densities [Hyvärinen, 2005], appearing e.g. in energy based models (EBM) [LeCun et al., 2006].

**Kernel Stein discrepancy**  To assess discrepancies between two probability distributions $p$ and $q$, the Stein discrepancy (w.r.t. class $\mathcal{B} \subset \mathcal{F}$) is defined as [Gorham and Mackey, 2015]

$$\mathrm{SD}(p\|q, \mathcal{B}) = \sup_{f \in \mathcal{B}}\{|\mathbb{E}_p[\mathcal{A}_q f] - \underbrace{\mathbb{E}_p[\mathcal{A}_p f]}_{=0}|\} = \sup_{f \in \mathcal{B}} |\mathbb{E}_p[\mathcal{A}_q f]|. \tag{3}$$

As the $\sup f$ over a general class $\mathcal{B}$ can be difficult to compute, taking $\mathcal{B}$ as the unit ball of a reproducing kernel Hilbert space (RKHS) has been suggested in Chwialkowski et al. [2016], Liu et al. [2016], resulting in the *kernel Stein discrepancy* (KSD) defined as

$$\mathrm{KSD}(p\|q, \mathcal{H}) = \sup_{f \in \mathcal{B}_1(\mathcal{H})} |\mathbb{E}_p[\mathcal{A}_q f]|. \tag{4}$$

Denoting by $k$ the reproducing kernel associated with the RKHS $\mathcal{H}$ over a set $\mathcal{X}$, the reproducing property ensures that $\forall f \in \mathcal{H}$, $f(\boldsymbol{x}) = \langle f, k(\boldsymbol{x}, \cdot)\rangle_{\mathcal{H}}, \forall \boldsymbol{x} \in \mathcal{X}$. Algebraic manipulations yield

$$\mathrm{KSD}^2(q\|p) = \mathbb{E}_{\boldsymbol{x}, \tilde{\boldsymbol{x}} \sim p}[u_q(\boldsymbol{x}, \tilde{\boldsymbol{x}})], \tag{5}$$

where $u_q(\boldsymbol{x}, \tilde{\boldsymbol{x}}) = \langle \mathcal{A}_q k(\boldsymbol{x}, \cdot), \mathcal{A}_q k(\tilde{\boldsymbol{x}}, \cdot)\rangle_{\mathcal{H}}$, which takes the exact $\sup$ without approximation and does not involve the (sample) distribution $p$. Then, $\mathrm{KSD}^2$ can be estimated through empirical means, over samples from $p$. For example, V-statistics [Van der Vaart, 2000] and U-statistics [Lee, 1990] estimates are

$$\mathrm{KSD}_v^2(q\|p) = \frac{1}{m^2} \sum_{i,j} u_q(\boldsymbol{x}_i, \boldsymbol{x}_j), \qquad \mathrm{KSD}_u^2(q\|p) = \frac{1}{m(m-1)} \sum_{i \neq j} u_q(\boldsymbol{x}_i\, \boldsymbol{x}_j). \tag{6}$$

KSD has been proposed as discrepancy measure between distributions for testing model goodness-of-fit in Chwialkowski et al. [2016] and Liu et al. [2016].

**KSD testing procedure**  Suppose we have observed samples $\boldsymbol{x}_1, \ldots, \boldsymbol{x}_n$ from the *unknown* distribution $p$. To test the null hypothesis $H_0 : p = q$ against the (broad class of) alternative hypothesis $H_1 : p \neq q$, KSD can be empirically estimated via Eq. 6. The null distribution is usually simulated via the wild-bootstrap procedure [Chwialkowski et al., 2014]. Then if the empirical quantile, i.e. the proportion of wild bootstrap samples that are larger than $\mathrm{KSD}_v^2(q\|p)$, is smaller than the pre-defined test level (or significance level) $\alpha$, the null hypothesis is rejected; otherwise the null hypothesis is not rejected. In this way, a systematic non-parametric goodness-of-fit testing procedure is obtained, which is applicable to unnormalised models.

## 3  Non-Parametric kernel Stein discrepancies

The construction of a KSD relies on the knowledge of the density model, up to normalisation. However, for deep generative models where the density function is not explicitly known, the computation of the Stein operator in Eq. 2, which is based on an explicit parametric density, is no longer feasible.

While in principle one could estimate the multivariate density function from synthetic data, density estimation in high dimensions is known to be problematic, see for example Scott and Sain [2005]. Instead, Stein's method allows to use a two-step approach: For data in $\mathbb{R}^m$, we first pick a coordinate $i \in [m] := \{1, \ldots, m\}$, and then we characterize the univariate conditional distribution of that coordinate, given the values of the other coordinates. Using score Stein operators from Ley et al. [2017], this approach only requires knowledge or estimation of univariate conditional score functions.

---

[1]also referred to as Langevin Stein operator [Barp et al., 2019].

First we introduce some notation. Observed data $\boldsymbol{z}_1, \ldots, \boldsymbol{z}_n$ are taken to be column vectors, $\boldsymbol{z}_i = (z_i^{(1)}, \ldots, z_i^{(m)})^\top \in \mathbb{R}^m$. For the generative model $G$ we write $\boldsymbol{X} \sim G$ to denote a random $\mathbb{R}^m$-valued element from the (often only given implicitly) distribution which is underlying $G$. Using $G$, we generate $N$ samples denoted by $\boldsymbol{y}_1, \ldots, \boldsymbol{y}_N$. In our case, $n$ is fixed and $n \ll N$, allowing $N \to \infty$ in theoretical results. The kernel of an RKHS is denoted by $k$ and is assumed to be bounded. For $\boldsymbol{x} \in \mathbb{R}^m$, $x \in \mathbb{R}$ and $g(\boldsymbol{x}) : \mathbb{R}^m \to \mathbb{R}$, we write $g_{x^{(-i)}}(x) : \mathbb{R} \to \mathbb{R}$ for the univariate function which acts only on the coordinate $i$ and fixes the other coordinates to equal $x^{(j)}, j \neq i$, so that $g_{x^{(-i)}}(x) = g(x^{(1)}, \ldots, x^{(i-1)}, x, x^{(i+1)}, \ldots, x^{(m)})$.

For $i \in [m]$ let $\mathcal{T}^{(i)}$ denote a Stein operator for the conditional distribution $Q^{(i)} = Q_{x^{(-i)}}^{(i)}$ with $\mathbb{E}_{Q_{x^{(-i)}}^{(i)}} g_{x^{(-i)}}(x) = \mathbb{E}[g_{y^{(-i)}}(Y)|Y^{(j)} = y^{(j)}, j \neq i]$. The proposed Stein operator $\mathcal{A}$ acting on functions $g : \mathbb{R}^m \to \mathbb{R}$ underlying the non-parametric Stein operator is

$$\mathcal{A}g(x^{(1)}, \ldots, x^{(m)}) = \frac{1}{m} \sum_{i=1}^{m} \mathcal{T}^{(i)} g_{x^{(-i)}}(x^{(i)}). \tag{7}$$

We note that for $\boldsymbol{X} \sim q$, the Stein identity $\mathbb{E}\mathcal{A}g(\boldsymbol{X}) = 0$ holds and thus $\mathcal{A}$ is a Stein operator. The domain of the operator will depend on the conditional distribution in question. Instead of using the weights $w_i = \frac{1}{m}$ for the $\mathcal{T}^{(i)}$, other positive weights which sum to 1 would be possible, but for simplicity we use equal weights. A more detailed theoretical justification of Eq. 7 is given in Appendix A.

In what follows we use as Stein operator for a differentiable univariate density $q$ the score operator from Eq. 2, given by

$$\mathcal{T}_q^{(i)} f(x) = f'(x) + f(x) \frac{q'(x)}{q(x)}. \tag{8}$$

In Proposition D.1 of Appendix D we shall see that the operator in Eq. 7 equals the score-Stein operator in Eq. 2; in Appendix D an example is also given. For the development in this paper, Eq. 7 is more convenient as it relates directly to conditional distributions. Other choices of Stein operators are discussed for example in Ley et al. [2017], Mijoule et al. [2021], Xu [2022].

**Re-sampling Stein operators**    The Stein operator Eq. 7 depends on all coordinates $i \in [m]$. When $m$ is large we can estimate this operator via re-sampling with replacement, as follows. We draw $B$ samples $\{i_1, \ldots, i_B\}$ with replacement from $[m]$ such that $\{i_1, \ldots, i_B\} \sim \text{Multinom}(B, \{\frac{1}{m}\}_{i \in [m]})$. The re-sampled Stein operator acting on $f : \mathbb{R}^m \to \mathbb{R}$ is

$$\mathcal{A}^B f(\boldsymbol{z}) := \frac{1}{B} \sum_{b=1}^{B} \mathcal{A}^{(i_b)} f(\boldsymbol{z}). \tag{9}$$

Then we have $\mathbb{E}\mathcal{A}^B f(\boldsymbol{X}) = \frac{1}{B} \sum_{b=1}^{B} \mathbb{E}\mathcal{A}^{(i_b)} f(\boldsymbol{X}) = 0$. So $\mathcal{A}^B$ is again a Stein operator.

In practice, when $m$ is large, the stochastic operator in Eq. 9 creates a computationally efficient way for comparing distributions. A similar re-sampling strategy for constructing stochastic operators has been suggested in the context of Bayesian inference [Gorham et al., 2020], where conditional score functions, which are given in parametric form, are re-sampled to derive score-based (or Langevin) Stein operators for posterior distributions. The conditional distribution in Stein operators has been considered for example in Singhal et al. [2019] and Wang et al. [2018]; see also Zhuo et al. [2018] and Liu and Wang [2016] in the context of graphical models. In graphical models, the conditional distribution is simplified to conditioning on the Markov blanket [Wang et al., 2018], which is a subset of the full coordinate; however, no random re-sampling is used. Conditional distributions also apply in message passing, but there, the sequence of updates is ordered.

**Estimating Stein operators via score matching**    Usually the score function $q'/q$ in Eq. 8 is not available but needs to be estimated. An efficient way of estimating the score function is through score-matching, see for example Hyvärinen [2005], Song and Kingma [2021], Wenliang et al. [2019]. Score matching relies on the following score-matching (SM) objective [Hyvärinen, 2005],

$$J(p\|q) = \mathbb{E}_p \left[ \|\nabla \log p(\boldsymbol{x}) - \nabla \log q(\boldsymbol{x})\|^2 \right], \tag{10}$$

---

**Algorithm 1** Estimating the conditional probability via summary statistics

---

**Input:** Generator $G$; summary statistics $t(\cdot)$; number of samples $N$ from $G$; re-sample size $B$
**Procedure:**
  1: Generate samples $\{\boldsymbol{y}_1, \ldots, \boldsymbol{y}_N\}$ from $G$.
  2: Generate coordinate index sample $\{i_1, \ldots, i_B\}$
  3: For $i_b \in [m], l \in [N]$, estimate $q(z^{(i_b)}|t(z^{-i_b})$ from samples $\{y_l^{(i_b)}, t(y_l^{-i_b})\}_{l \in [N]}$ via the score-matching objective in Eq. 10.
**Output:** $\widehat{s}_{t,N}^{(i)}(z^{(i)}|t(z^{(-i)})), \forall i \in [m]$.

---

which is particularly useful for unnormalised models such as EBMs. Additional details are included in Appendix E. Often score matching estimators can be shown to be consistent, see for example Song et al. [2020]. Proposition 3.1, proven in Appendix B, gives theoretical guarantees for the consistency of a general form of Stein operator estimation, as follows.

**Proposition 3.1.** *Suppose that for $i \in [m]$, $\widehat{s}_N^{(i)}$ is a consistent estimator of the univariate score function $s^{(i)}$. Let $\mathcal{T}^{(i)}$ be a Stein operator for the univariate differentiable probability distribution $Q^{(i)}$ of the generalised density operator form Eq. 8. Let*

$$\widehat{\mathcal{T}}_N^{(i)}g(x) = g'(x) + g(x)\widehat{s}_N^{(i)} \qquad \text{and} \qquad \widehat{\mathcal{A}}g = \frac{1}{m}\sum_i \widehat{\mathcal{T}}_N^{(i)}g_{x^{(-i)}}.$$

*Then $\widehat{\mathcal{T}}_N^{(i)}$ is a consistent estimator for $\mathcal{T}^{(i)}$, and $\widehat{\mathcal{A}}$ is a consistent estimator of $\mathcal{A}$.*

**Non-parametric Stein operators with summary statistics** In practice, the data $y^{(-i)} \in \mathbb{R}^{m-1}$ can be high dimensional, e.g. image pixels, and the observations can be sparse. Thus, estimation of the conditional distribution can be unstable or exponentially large sample size is required. Motivated by Xu and Reinert [2021] and Xu and Reinert [2022], we use a low-dimensional measurable non-trivial summary statistic $t$ and the conditional distribution of the data given $t$ as new target distributions. Heuristically, if two distributions match, then so do their conditional distributions. Thus, the conditional distribution $Q^{(i)}(A)$ is replaced by $Q_t^{(i)}(A) = \mathbb{P}(X^{(i)} \in A|t(x^{(-i)}))$. Setting $t(x^{(-i)}) = x^{(-i)}$ replicates the actual conditional distribution. We denote the univariate score function of $q_t(x|t(x^{(-i)}))$ by $s_t^{(i)}(x|t(x^{(-i)}))$, or by $s_t^{(i)}(x)$ when the context is clear. The summary statistics $t(x^{(-i)})$ can be univariate or multivariate, and they may attempt to capture useful distributional features. Here we consider univariate summary statistics such as the sample mean.

Taking these component-dependent conditional Stein operators results in the Stein operator in general to be no longer equal to the Langevin Stein operator of the corresponding conditional multivariate distribution. Thus, Proposition D.1 (presented in the Appendix) no longer applies.

The non-parametric Stein operator enables the construction of Stein-based statistics based on Eq. 7 with estimated score functions $\widehat{s}_{t,N}^{(i)}$ using generated samples from the model $G$, as shown in Algorithm 1. The re-sampled non-parametric Stein operator is

$$\widehat{\mathcal{A}_{t,N}^B}g = \frac{1}{B}\sum_b \widehat{\mathcal{T}}_{t,N}^{(i_b)}g_{x^{(-i_b)}} = \frac{1}{B}\sum_b \left(g'_{x^{(-i_b)}} + g_{x^{(-i_b)}}\widehat{s}_{t,N}^{(i)}\right). \tag{11}$$

**Non-parametric kernel Stein discrepancy** Using the non-parametric Stein operator in Eq. 11 we define the corresponding non-parametric Stein discrepancy (NP-KSD) employing the Stein discrepancy notion in Eq. 3 and choosing as set of test functions the unit ball of the RKHS. Similarly to Eq. 4, we define the NP-KSD with summary statistic $t$ as

$$\text{NP-KSD}_t(G\|p) = \sup_{f \in \mathcal{B}_1(\mathcal{H})} \mathbb{E}_p[\widehat{\mathcal{A}}_{t,N}^B f]. \tag{12}$$

A similar quadratic form as in Eq. 5 applies to give

$$\text{NP-KSD}_t^2(G\|p) = \mathbb{E}_{\boldsymbol{x}, \tilde{\boldsymbol{x}} \sim p}[\widehat{u}_{t,N}^B(\boldsymbol{x}, \tilde{\boldsymbol{x}})], \tag{13}$$

where $\widehat{u}_{t,N}^B(\boldsymbol{x}, \tilde{\boldsymbol{x}}) = \langle \widehat{\mathcal{A}}_{t,N}^B k(\boldsymbol{x}, \cdot), \widehat{\mathcal{A}}_{t,N}^B k(\tilde{\boldsymbol{x}}, \cdot) \rangle_{\mathcal{H}}$. The empirical estimate is

$$\widehat{\text{NP-KSD}}_t^2(G\|p) = \frac{1}{n^2} \sum_{i,j \in [n]} [\widehat{u}_{t,N}^B(\boldsymbol{z}_i, \boldsymbol{z}_j)], \tag{14}$$

where $\mathbb{S} = \{\boldsymbol{z}_1, \ldots, \boldsymbol{z}_n\} \sim p$. Thus, NP-KSD allows the computation between a set of samples and a generative model, enabling the quality assessment of synthetic data generators for implicit models.

The relationship between NP-KSD and KSD is clarified in the following result; we use the notation $\hat{\mathbf{s}}_{t,N} = (\hat{s}_{t,N}(x^{(i)}), i \in [m])$. Here we set

$$\text{KSD}_t^2(q_t\|p) = \mathbb{E}_{\boldsymbol{x}, \tilde{\boldsymbol{x}} \sim p}[\langle \mathcal{A}_t k(\boldsymbol{x}, \cdot), \mathcal{A}_t k(\tilde{\boldsymbol{x}}, \cdot) \rangle_{\mathcal{H}} \quad \text{with} \quad \mathcal{A}_t g(\boldsymbol{x}) := \frac{1}{m} \sum_{i=1}^m \mathcal{T}_{q_t}^{(i)} g_{x^{(-i)}}(x^{(i)})$$

$$\tag{15}$$

as in Eq. 7, and following Eq. 8, $\mathcal{T}_{q_t}^{(i)} g_{x^{(-i)}}(x) = g'_{x^{(-i)}}(x) + g_{x^{(-i)}}(x) s_t^{(i)}(x|t(x^{(-i)}))$. More details about the interpretation of this quantity are given in Appendix B.1.

**Theorem 3.2.** *Assume that the score function estimator vector $\hat{\mathbf{s}}_{t,N} = (\hat{s}_{t,N}^{(i)}, i = 1, \ldots, m)^\top$ is asymptotically normal with mean $0$ and covariance matrix $N^{-1}\Sigma_s$. Then $\text{NP-KSD}_t^2(G\|p)$ converges in probability to $\text{KSD}_t^2(q_t\|p)$ at rate at least $\min(B^{-\frac{1}{2}}, N^{-\frac{1}{2}})$.*

The proof of Theorem 3.2, which is found in Appendix B, also shows that the distribution $\text{NP-KSD}_t^2(G\|p) - \text{KSD}_t^2(q_t\|p)$ involves a mixture of normal variables. The assumption of asymptotic normality for score matching estimators is often satisfied, see for example Song et al. [2020].

**Model assessment with NP-KSD**  Given an implicit generative model $G$ and a set of observed samples $\mathbb{S} = \{\boldsymbol{z}_1, \ldots, \boldsymbol{z}_n\}$, we aim to assess whether the observed samples $\mathbb{S}$ are consistent with the hypothesis that they were generated from $G$. This test assumes that samples generated from $G$ follow some (unknown) distribution $q$ and that the samples $\mathbb{S}$ are generated according to some (unknown) distribution $p$. The null hypothesis is $H_0 : p = q$ while the alternative is $H_1 : p \neq q$. We note that the observed sample size $n$ is fixed.

**NP-KSD testing procedures**  NP-KSD can be applied for testing the above hypothesis using the testing procedure outlined in Algorithm 2. In contrast to the KSD testing procedure in Section 2, the NP-KSD test in Algorithm 2 is a Monte Carlo based test [Xu and Reinert, 2021, 2022, Schrab et al., 2022] for which the null distribution is approximated via samples generated from $G$ instead of the wild bootstrap procedure as in Chwialkowski et al. [2014]. The reasons for employing the Monte Carlo testing strategy instead of the wild-bootstrap are 1). The non-parametric Stein operator depends on the random function $\hat{s}_t$ so that classical results for V-statistics convergence which assume that the sole source of randomness is the bootstrap may not apply[2]. 2). While the wild-bootstrap is asymptotically consistent as observed sample size $n \to \infty$, it may not necessarily control the type-I error in a non-asymptotic regime where $n$ is fixed. More details can be found in Appendix F.1.

Here we note that any test which is based on the summary statistic $t$ will only be able to test for a distribution up to equivalence of their distributions with respect to the summary statistic $t$; two distributions $P$ and $Q$ are equivalent w.r.t. the summary statistics $t$ if $P(\boldsymbol{X}|t(\boldsymbol{X})) = Q(\boldsymbol{X}|t(\boldsymbol{X}))$. Thus the null hypothesis for the NP-KSD test is that the distribution is equivalent to $P$ with respect to $t$. Hence, the null hypothesis specifies the conditional distribution, not the unconditional distribution.

**Related works**  To assess whether an implicit generative models can generate samples that are *significantly* consistent with the desired data model, several hypothesis testing procedures have been studied. Jitkrittum et al. [2018] has proposed two kernel-based tests, called Relative Unbiased Mean Embedding (Rel-UME) test and Relative Finite-Set Stein Discrepancy (Rel-FSSD) test, for testing relative model goodness-of-fit, that is, testing whether model S is a better fit to the observations than model R. While Rel-UME is applicable to implicit generative models, Rel-FSSD still requires explicit knowledge of the unnormalised density.

The idea for assessing sample quality for implicit generative models could be viewed as a very imbalanced two-sample problem, where samples generated from the implicit model are compared

---

[2]A KSD with random Stein kernel has been briefly discussed in Fernández et al. [2020] when the $h_q$ function requires estimation from relevant survival functions.

---

**Algorithm 2** Assessment procedures for implicit generative models

---

**Input:** Observed sample set $\mathbb{S} = \{z_1, \ldots, z_n\}$; the generator $G$ and generated sample size $N$; estimation statistics $t$; RKHS kernel $K$; re-sampling size $B$; bootstrap sample size $b$; confidence level $\alpha$;

1: Estimate $\widehat{s}(z^{(i)}|t(z^{(-i)}))$ based on Algorithm 1.
2: Uniformly generate re-sampling index $\{i_1, \ldots, i_B\}$ from $[m]$, with replacement.
3: Compute $\tau = \widehat{\text{NP-KSD}}^2(\widehat{s}_t; \mathbb{S})$ in Eq. (14).
4: Simulate $\mathbb{S}_i = \{y'_1, \ldots, y'_n\}$ for $i \in [b]$ from $G$.
5: Compute $\tau_i = \widehat{\text{NP-KSD}}^2(\widehat{s}_t; \mathbb{S}_i)$ in again with index re-sampling.
6: Estimate the empirical (1-$\alpha$) quantile $\gamma_{1-\alpha}$ via $\{\tau_1, \ldots, \tau_b\}$.

**Output:** Reject the null hypothesis if $\tau > \gamma_{1-\alpha}$; otherwise do not reject.

---

with the observed data. For general two-sample problems, maximum-mean-discrepancy (MMD) could in principle also be applied for assessing sample qualities for implicit models. With an efficient choice of (deep) kernel, Liu et al. [2020] applied MMD tests to assess the distributional difference for image data, e.g. MNIST [LeCun et al., 1998] v.s. digits image trained via deep convolutional GAN (DCGAN) [Radford et al., 2015]; CIFAR10 [Krizhevsky, 2009] v.s. CIFAR10.1 [Recht et al., 2019]. However, we note that the MMD-based two-sample approach for model assessment suffers from two major limitations. Firstly, MMD tests [Gretton et al., 2012a] require a balanced sample size for two sets of samples; if $n_1$ and $n_2$ are the sizes for each sample set, then MMD tests have well-controlled type-I error when $n_1 \approx n_2$. In the model assessment setting, we may observe a small fixed sample set $n$ while being able to generate $N \gg n$ samples from the generator. The second limitation concerns the idea to use matched sample size in order to enjoy the theoretical guarantees which are available for MMD tests, in which case we may only generate $O(n)$ samples from the generator to balance the observed sample size. As such, the empirical estimator for MMD becomes less accurate, which would significantly reduce the test power for small $n$. This approach would not exploit the fact that the data generator can generate a very large amount of samples if desired. Detailed discussions on MMD two-sample testing procedures can be found in Appendix F.1, including Table 5 showing uncontrolled type-I errors for imbalanced sample size.

## 4 Experiments

### 4.1 Baseline and competing approaches

We illustrate the proposed NP-KSD testing procedure with different choice of summary statistics. We denote by **NP-KSD** the version which uses the estimation of the conditional score, i.e. $t(x^{(-i)}) = x^{(-i)}$; by **NP-KSD_mean** the version which uses conditioning on the mean statistics, i.e. $t(x^{(-i)}) = \frac{1}{m-1}\sum_{j \neq i} x^{(j)}$; and by **NP-KSD_G** the version which fits a Gaussian model as conditional density[3].

Two-sample testing methods can be useful for model assessment, where the observed sample set is tested against sample set generated from the model. In our setting where $n \ll N$, we consider a consistent non-asymptotic MMD-based test, **MMDAgg** [Schrab et al., 2021], as our competing approach; see Appendix F.1 for details on computing MMDAgg statistics and discussions on its comparison with the MMD tests [Gretton et al., 2012a][4].

For synthetic distributions where the null models have explicit densities, we include the **KSD** goodness-of-fit testing procedure in Section 2 as the baseline. Gaussian kernels are used and the median heuristic [Gretton et al., 2007] is applied for bandwidth selection. As a caveat, in view of [Gorham and Mackey, 2015], when the kernel decays more rapidly than the score function grows,

---

[3]**NP-KSD_G** for non-Gaussian densities is generally mis-specified. We deliberately check this case to assess the robustness of the NP-KSD procedure under model mis-specification.

[4]We remark on another desirable property for MMDAgg. MMDAgg is able to (implicitly) select the optimal kernel parameter to achieve better test power without *data-splitting*, where data-splitting results into smaller sample size available for performing the test, thus reducing data efficiency. However, this property is not the reason we choose MMDAgg as our baseline here; we choose it because MMDAgg is a non-asymptotic MMD-based test which is capable of handling the an imbalanced sample situation, i.e. $n \ll N$.

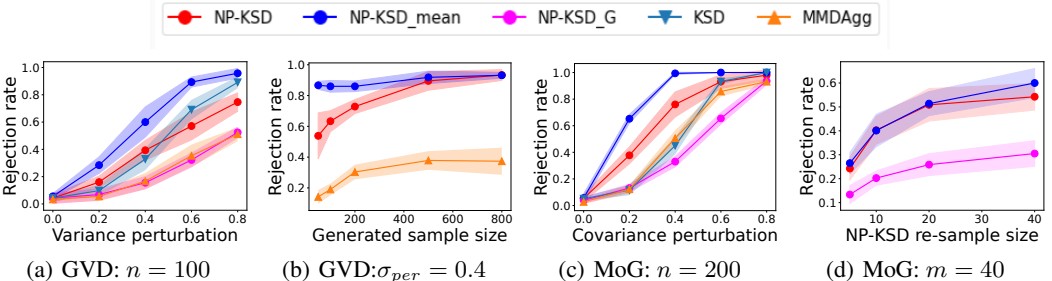

Figure 1: Rejection rates of the synthetic distributions: test level $\alpha = 0.05$; 100 trials per round of experiment; 10 rounds of experiment are taken for average and standard deviation; bootstrap sample size $b = 500$; $m = 3$ for (a) and (b); $m = 6$ for (c); $n = 100$, $\sigma_{per} = 0.5$ for (d).

then identifiability of $q_t$ through a KSD method may not be guaranteed. We note that MMD tests do not have controlled type-I error when $n \ll N$, thus are not suitable in this setting. While MMD-based methods to compare and criticise two generative models have been explored [Lloyd and Ghahramani, 2015, Sutherland et al., 2017], MMD tests [Gretton et al., 2012a] are therefore not included in the comparison list. Detailed discussions and illustrations are found in Appendix F.1.

## 4.2 Experiments on synthetic distributions

**Gaussian Variance Difference (GVD)** We first consider a standard synthetic setting, studied in Jitkrittum et al. [2017], in which the null distribution is multivariate Gaussian with mean zero and identity covariance matrix. The alternative is set to perturb the the diagonal terms of the covariance matrix, i.e. the variances, all by the same amount.

The rejection rate against the variances perturbation is shown in Figure 1(a). From the result, we see that all the tests presented have controlled type-I error. For all the tests the power increases with increased perturbation. **NP-KSD** and **NP-KSD_mean** outperform the **MMDAgg** approach. Using the mean statistics, **NP-KSD_mean** is having slightly higher power than **KSD**. The mis-specified **NP-KSD_G** has lower power, but is still competitive to **MMDAgg**.

The test power against the sample size $N$ generated from the null model is shown in Figure 1(b). The generated samples are used as another sample set for the **MMDAgg** two-sample procedure, while used for estimating the conditional score for NP-KSD-based methods. As the generated sample size increases, the power of **MMDAgg** increases more slowly than that of the NP-KSD-based methods, which achieve maximum test power in the presented setting. The NP-KSD-based tests tend to have lower variability of the test power, indicating more reliable testing procedures than **MMDAgg**.

**Mixture of Gaussian (MoG)** Next, we consider as a more difficult problem that the null model is a two-component mixture of two independent Gaussians. Both Gaussian components have identity covariance matrix. The alternative is set to perturb the covariance between adjacent coordinates.

The rejection rates against this perturbation of covariance terms are presented in Figure 1(c). The results show controlled type-I error. The **NP-KSD** and **NP-KSD_mean** tests have better test power compared to **KSD** and **MMDAgg**, although **NP-KSD** has slightly higher variance. Among the NP-KSD tests, the smallest variability is achieved by **NP-KSD_mean**. For the test with $m = 40$, we also vary the re-sample size $B$. As shown in Figure 1(d), the variability of the average test power increased slightly. From the result, we also see that for $B = 20 = m/2$ the test power is already competive compared to $B = 40$. Additional experimental results including computational runtime and training generative models for synthetic distributions are included in Appendix C.

## 4.3 Applications to deep generative models

For real-world applications, we assess models trained from well-studied generative modelling procedures, including a Generative Adversarial Network (**GAN**) [Goodfellow et al., 2014] with multilayer perceptron (MLP), a Deep Convolutional Generative Adversarial Network (**DCGAN**) [Radford et al., 2015], and a Variational Autoencoder (**VAE**) [Kingma and Welling, 2013]. We also consider a Noise

|         | GAN_MLP | DCGAN | VAE  | NCSN | Real |
|---------|---------|-------|------|------|------|
| NP-KSD   | 1.00    | 0.92  | 1.00 | 1.00 | 0.03 |
| NP-KSD_m | 1.00    | 1.00  | 1.00 | 1.00 | 0.01 |
| MMDAgg   | 1.00    | 0.73  | 0.93 | 1.00 | 0.06 |

Table 1: Rejection rate for MNIST generative models.

|         | DCGAN | NCSN | CIFAR10.1 | Real |
|---------|-------|------|-----------|------|
| NP-KSD   | 0.68  | 0.73 | 0.92      | 0.06 |
| NP-KSD_m | 0.74  | 0.81 | 0.96      | 0.02 |
| MMDAgg   | 0.48  | 0.57 | 0.83      | 0.07 |

Table 2: Rejection rate for CIFAR10 generative models.

Conditional Score Network (**NCSN**) [Song and Ermon, 2020], which is a score-based generative modelling approach, where the score functions are learned [Song and Ermon, 2019] to performed annealed Langevin dynamics for sample generation. We also denote **Real** as the scheme that generates samples randomly from the training data, by using a held-out set of samples from the real data, and which hence can be viewed as an implicit generator of the null distribution.

**MNIST Dataset** This dataset contains $28 \times 28$ grey-scale images of handwritten digits [LeCun et al., 1998][5]. It consist of $60,000$ training samples and $10,000$ test samples. Deep generative models in Table 1 are trained using the training samples. We assess the quality of these trained generative models by testing against the true observed MNIST samples (from the test set). Samples from both distributions are visually illustrated in Figure 3 in Appendix C. For our experiments, $600$ samples are generated from the generative models and $100$ samples are used for the test; the samples are then down-sampled into $7 \times 7$ images, as in Schrab et al. [2021]. The test level is $\alpha = 0.05$. From Table 1, we see that all the deep generative models have high rejection rates. Testing with the **Real** scheme has controlled type-I error. Thus, NP-KSD detects that the "real" data are a true sample set from the underlying dataset, but often rejects the synthetic data generators.

**CIFAR10 Dataset** This dataset contains $32 \times 32$ RGB coloured images [Krizhevsky, 2009][6]. It consist of $50,000$ training samples and $10,000$ test samples. Deep generative models in Table 2 are trained using the training samples, and test samples are randomly drawn from the test set. Samples are illustrated in Figure 4 in Appendix C. We also compare with the CIFAR10.1 dataset [Recht et al., 2018][7], which is created to differ from CIFAR10 to investigate generalisation power of training classifiers. For our experiments, $800$ samples are generated from the generative models and $200$ samples are used for the test; the samples are then down-sampled into $8 \times 8$ images, in a similar fashion as in Schrab et al. [2021]. The test level is $\alpha = 0.05$. Table 2 shows higher rejection rates for NP-KSD tests compared to MMDAgg, echoing the results for synthetic distributions. The trained **DCGAN** generates samples with lower rejection rate in the CIFAR10 dataset than in the CIFAR10.1 dataset. We also see that the score-based NCSN has higher rejection rate than the non-score-based DCGAN, despite NP-KSD being a score-based test. The distribution difference between CIFAR10 and CIFAR10.1 can be well-distinguished from the tests. Testing with the **Real** scheme again has controlled type-I error.

## 5   Conclusion and future directions

Synthetic data are in high demand, for example for training ML procedures; quality is important. Synthetic data which miss important features in the data can lead to erroneous conclusions, which for example in medical applications could be fatal, and in loan applications could be detrimental to personal or business development. NP-KSD provides a method for assessing synthetic data generators

---

[5]`https://pytorch.org/vision/main/generated/torchvision.datasets.MNIST.html`
[6]`https://pytorch.org/vision/stable/generated/torchvision.datasets.CIFAR10.html`
[7]`https://github.com/modestyachts/CIFAR-10.1/tree/master/datasets`

which comes with theoretical guarantees. Our experiments on synthetic data have shown that NP-KSD achieves good test power and controlled type-I error. On real data, NP-KSD detects samples from the true dataset. That none of the classical deep learning methods used in this paper has a satisfactory rejection rate indicates scope for further developments in synthetic data generation.

Future research will assess alternatives to the computer-intensive Monte Carlo method for estimating the null distribution, for example through adapting wild-bootstrap procedures. It will explore alternative choices of score estimation as well as of kernel functions.

Finally, some caution is advised. The choice of summary statistic may have strong influence on the results and a classification based on NP-KSD may still miss some features. Erroneous decisions could be reached when training classifiers. Without scrutiny this could lead to severe consequences for example in health science applications. Future work will devote more attention on analysing the choice of summary statistic. Yet NP-KSD is an important step towards understanding black-box data generating methods and thus understanding their potential shortcomings.

## Acknowledgements

The authors thank the anonymous reviewers for their constructive comments and suggestions to improve the paper. G.R. and W.X. acknowledge the support from EPSRC grant EP/T018445/1. G.R is also supported in part by EPSRC grants EP/W037211/1, EP/V056883/1, and EP/R018472/1.

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
