# Supplementary Information for A Kernelised Stein Statistic for Assessing Implicit Generative Models

## A   Justification of the Stein operator

Here we justify the two-step approach for constructing a Stein operator.

### A.1   Step 1: A non-parametric Stein operator

Suppose we can estimate the conditional distribution from data. Then we can create a Markov chain with values in $(\mathbb{R}^d)^m$ as follows. Starting with $Z_0 = \{x_1, \ldots, x_m\}$ with $x_i \in \mathbb{R}^d$ for $i = 1, \ldots, m$ (often we choose $d = 1$), we pick an index $I \in [m]$ at random. If $I = i$ we replace $x_i$ by $X_i'$ drawn from the conditional distribution of $X_i$ given $(X_j : j \neq i)$. This gives $Z_1 = (x_1, \ldots, x_{i-1}, X_i', x_{i+1}, \ldots, x_m)$[8] see for example Reinert [2005]. To make this a continuous-time Markov process generator, we wait an exponential(1)-distributed time before every change.

This generator induces a Stein operator for the target distribution as follows. Here we take $d = 1$ for clarity; the generalisation to other $d$ is straightforward. Let $f : \mathbb{R}^m \to \mathbb{R}$ and consider the expectation w.r.t. the one-step evolution of the Markov chain

$$\mathbb{E}_{-i}[f(x^{(1)}, \ldots, x^{(i-1)}, X^{(i)}, x^{(i+1)}, \ldots, x^{(m)})]$$
$$= \int f(x^{(1)}, \ldots, x^{(i-1)}, y, x^{(i+1)}, \ldots, x^{(m)}) \mathbb{P}(X^{(i)} = y | X^{(j)} = x^{(j)}, j \neq i).$$

We now consider the index $i$ as the $i$-th coordinate of multivariate random variables in $\mathbb{R}^m$. The conditional expectation here, fixing all but the $i$-th *coordinate* term, only depends on the univariate conditional distribution $Q^{(i)}$ given by $Q^{(i)}(A) = \mathbb{P}(X^{(i)} \in A | X^{(j)} = x^{(j)}, j \neq i)$. Thus, the Stein operator induced from the Markov chain has the form

$$\mathcal{A}f(z) = \mathcal{A}^{(I)}f(z) \tag{16}$$

where

$$\mathcal{A}^{(i)}f(\boldsymbol{x}) = \mathbb{E}_{-i}[f(x^{(1)}, \ldots, x^{(i-1)}, X^{(i)}, x^{(i+1)}, \ldots, x^{(m)})] - f(\boldsymbol{x}). \tag{17}$$

From the law of total expectation it follows that the Stein identity holds.

### A.2   Step 2: marginal Stein operators

In Eq. (17), the expectation

$$\mathbb{E}_{-i}[f(x^{(i)}, \ldots, x^{(i-1)}, X^{(i)}, x^{(i+1)}, \ldots, x^{(m)})] - f(x^{(1)}, \ldots, x^{(m)})$$

can itself be treated via Stein's method, by finding a Stein operator $\mathcal{T}^{(i)}$ and a function $g$ such that $g = g_f$ solves the $\mathcal{T}^{(i)}$-Stein equation Eq. (1) for $f$;

$$\mathcal{T}^{(i)}g(x) = \mathbb{E}_{-i}[f(x^{(1)}, \ldots, x^{(i-1)}, X^{(i)}, x^{(i+1)}, \ldots, x^{(m)})] - f(x^{(1)}, \ldots, x^{(m)}). \tag{18}$$

Fixing $x_j, j \neq i$ and setting $f^{(i)}(x) = f(x^{(1)}, \ldots, x^{(i-1)}, x, x^{(i+1)}, \ldots, x^{(m)})$, we view $\mathcal{T}^{(i)}$ as a Stein operator for a univariate distribution, acting on functions $g = g_{x^{(-i)}} : \mathbb{R} \to \mathbb{R}$.

Summarising the approach, the Stein operator $\mathcal{A}$ acting on functions $f : \mathbb{R}^m \to \mathbb{R}$ underlying the non-parametric Stein operator is

$$\mathcal{A}f(x^{(1)}, \ldots, x^{(m)}) = \mathcal{T}^{(I)}g_{f,x^{-I}}(x^{(I)}) \tag{19}$$

where $I \in [m]$ is a randomly chosen index. In view of Eq. (19) we take $g : \mathbb{R}^m \to \mathbb{R}$, write $g_{x^{(-i)}}(x) : \mathbb{R} \to \mathbb{R}$ for the univariate function which acts only on the coordinate $i$ and fixes the other

---

[8]Denote $Z_1 = (x^{(1)}, \ldots, x^{(i-1)}, X^{(i)'}, x^{(i+1)}, \ldots, x^{(m)}) \in \mathbb{R}^m$ where the superscript $(i)$ is used for the coordinate index.

coordinates to equal $x^{(-i)}$, we as Stein operator (using the same letter $\mathcal{A}$ as before, which is abuse of notation);

$$\mathcal{A}g(x^{(1)}, \ldots, x^{(m)}) = \mathcal{T}^{(I)} g_{x^{-I}}(x^{(I)}).$$

This formulation simplifies Eq. (19) in that we no longer have to consider the connection between $f$ and $g$. The final step is to note that when we condition on the random index $I$, again a Stein operator is obtained, as follows. As

$$\mathbb{E}_I[\mathcal{A}g(x^{(1)}, \ldots, x^{(m)})] = \frac{1}{m} \sum_{i=1}^{m} \mathcal{T}^{(i)} g_{x^{(-i)}}(x^{(i)}). \tag{20}$$

As $\mathbb{E}[\mathcal{T}^{(i)} g_{X^{(-i)}}(X^{(i)})] = 0$, the Stein identity is satisfied. The operator in Eq. (20) is the Stein operator given in Eq. (7).

The strategy of averaging over all coordinate terms $i \in [m]$ has also studied in variational inference, via coordinate ascent variational inference (CAVI); see for example Bishop and Nasrabadi [2006] which focuses on latent variable inference.

## B    Proofs and additional results

Assuming that if $f \in \mathcal{H}$ then $-f \in \mathcal{H}$ we can assume that the supremum over the expectation is non-negative, and with Eq. 12,

$$
\begin{aligned}
0 \leq \text{NP-KSD}_t(P\|Q) \quad &= \quad \sup_{f \in \mathcal{B}_1(\mathcal{H})} \mathbb{E}_p[\widehat{\mathcal{A}}_{t,N}^B f] \\
&= \quad \sup_{f \in \mathcal{B}_1(\mathcal{H})} \{\mathbb{E}_p \mathcal{A}_t f + \mathbb{E}_p[\widehat{\mathcal{A}}_{t,N}^B - \mathcal{A}_t]f\} \\
&= \quad \sup_{f \in \mathcal{B}_1(\mathcal{H})} \{\mathbb{E}_p \mathcal{A}_t f + \mathbb{E}_p[\widehat{\mathcal{A}}_{t,N}^B - \widehat{\mathcal{A}}_{t,N}]f + \mathbb{E}_p[f(\widehat{s}_{t,N}^{(i)} - \mathbf{s}_t)]\}. \tag{21}
\end{aligned}
$$

Here $\widehat{\mathcal{A}}_{t,N}$ is the Stein operator using the estimated conditional score function $\hat{s}_{t,N}$ with the estimation based on $N$ synthetic observations. We now assess the contribution to Eq. 21 which stems from estimating the score function. Note that here we only need to estimate a one-dimensional score function and hence the pitfalls of score estimation in high dimensions do not apply. We note however the contribution of Zhou et al. [2020] for a general framework.

Assume that we estimate the univariate conditional density $q_t^{(i)}$ based on $N$ samples. We assume that $q_t^{(i)}$ is differentiable, and we denote its score function by

$$s_t^{(i)}(x) = \frac{(q_t^{(i)})'(x)}{q_t^{(i)}(x)}.$$

We next prove an extension of Proposition 3.1.

**Proposition B.1.** *Suppose that for $i \in [m]$, $\widehat{s}_N^{(i)}$ is a consistent estimator of the univariate score function $s^{(i)}$. Let $\mathcal{T}^{(i)}$ be a Stein operator for the univariate differentiable probability distribution $Q^{(i)}$ of the generalised density operator form Eq. (8). Let*

$$
\begin{aligned}
\widehat{\mathcal{T}}_N^{(i)} g(x) \quad &= \quad g'(x) + g(x)\widehat{s}_N^{(i)} \\
\widehat{\mathcal{A}}g(\boldsymbol{x}) \quad &= \quad \widehat{\mathcal{T}}_N^{(I)} g_{x^{(-I)}}(x^{(I)}) \qquad and \\
\widehat{\mathcal{A}}_N g(\boldsymbol{x}) \quad &= \quad \frac{1}{m} \sum_{i \in [m]} \widehat{\mathcal{T}}_N^{(i)} g_{x^{(-i)}}(x^{(i)}).
\end{aligned}
$$

*Then $\widehat{\mathcal{T}}_N^{(i)}$ is a consistent estimator for $\mathcal{T}^{(i)}$, and $\widehat{\mathcal{A}}$ as well as $\widehat{\mathcal{A}}_N$ are consistent estimators of $\mathcal{A}$.*

*Proof.* Take a fixed $x$. As $\widehat{s}_N^{(i)}$ is a consistent estimator of $s^{(i)}$, it holds that for any $\epsilon > 0$ and for any $x$ in the range of $s^{(i)}$,

$$\mathbb{P}(|\widehat{s}_N^{(i)}(x) - s^{(i)}(x)| > \epsilon) \to 0$$

as $N \to \infty$. Here $\omega$ denotes the random element for the estimation, which is implicit in $\widehat{q}_N$. On the set

$$A_\epsilon = \left\{ |\widehat{s}_N^{(i)}(x) - s^{(i)}(x)| \leq \epsilon \right\}$$

we have that

$$|\widehat{\mathcal{T}}_N^{(i)} g(x) - \mathcal{T}^{(i)} g(x)| \leq \epsilon f(x).$$

For every fixed $x$ this expression tends to 0 as $\epsilon \to 0$. Hence consistency of $\widehat{T}_N$ follows. The last two assertions follow immediately from Eq. 19 and Eq. 20. $\qquad\square$

## B.1 Asymptotic behaviour of NP-KSD

Here we assess the asymptotic behaviour of NP-KSD$^2$. With $s_t$ denoting the conditional score function,

$$\text{NP-KSD}_t^2(G\|p) = \mathbb{E}_{\boldsymbol{x},\boldsymbol{x}' \sim p} \left\langle \mathcal{A}_{Q_t} k(\boldsymbol{x}, \cdot), \mathcal{A}_{Q_t} k(\boldsymbol{x}', \cdot) \right\rangle_{\mathcal{H}}$$

where $\mathcal{A}_{Q_t} k(\boldsymbol{x}, \cdot) = \mathcal{A}_t k(\boldsymbol{x}, \cdot)$ can be written as

$$
\begin{aligned}
\mathcal{A}_t k(\boldsymbol{x}, \cdot) &= \frac{1}{m} \sum_{i \in [m]} \left\{ \mathcal{A}_{\widehat{Q}_t^{(i)}} k(\boldsymbol{x}, \cdot) + k(\boldsymbol{x}, \cdot)(\widehat{s}_{t,N}^{(i)} - s_t^{(i)}) \right\} \\
&= \frac{1}{m} \sum_{i \in [m]} \left\{ \frac{\partial}{\partial x^{(i)}} k(\boldsymbol{x}, \cdot) + k(\boldsymbol{x}, \cdot) \widehat{s}_{t,N}^{(i)} + k(\boldsymbol{x}, \cdot)(\widehat{s}_{t,N}^{(i)} - s_t^{(i)}) \right\}. \quad (22)
\end{aligned}
$$

Recall that KSD$^2$ is given in Eq. (5) by

$$\text{KSD}^2(q\|p) = \mathbb{E}_{\boldsymbol{x}, \tilde{\boldsymbol{x}} \sim p}[\langle \mathcal{A}_q k(\boldsymbol{x}, \cdot), \mathcal{A}_q k(\tilde{\boldsymbol{x}}, \cdot) \rangle_{\mathcal{H}}],$$

where $\text{KSD}(q\|p)$ is a deterministic quantity which under weak assumption vanishes when $p = q$. Moreover,

$$\text{KSD}_t^2(q_t\|p) = \mathbb{E}_{\boldsymbol{x}, \tilde{\boldsymbol{x}} \sim p}[\langle \mathcal{A}_t k(\boldsymbol{x}, \cdot), \widehat{\mathcal{A}}_t k(\tilde{\boldsymbol{x}}, \cdot) \rangle_{\mathcal{H}}.$$

Disentangling this expression in general is carried out using Eq. (7).

*Remark* B.2. For a Gaussian kernel $k = k_G$ as used in this paper, we can exploit its factorisation;

$$k_G(\boldsymbol{x}, \tilde{\boldsymbol{x}}) = \exp\left\{ -\frac{1}{2\sigma^2} \sum_{i=1}^m \left( x^{(i)} - \tilde{x}^{(i)} \right)^2 \right\} = \prod_{i=1}^m \exp\left\{ -\frac{1}{2\sigma^2} \left( x^{(i)} - \tilde{x}^{(i)} \right)^2 \right\}.$$

In this situation, taking $g_{\boldsymbol{x}}(\cdot) = k_G(\boldsymbol{x}, \cdot)$, with $\cdot$ denoting an element in $\mathbb{R}^m$, gives

$$g_{\boldsymbol{x};x^{(-i)}}(\cdot) = \exp\left\{ -\frac{1}{2\sigma^2} \sum_{j:j \neq i}^m \left( x^{(j)} - (\cdot)^{(j)} \right)^2 \right\} \exp\left\{ -\frac{1}{2\sigma^2} \left( x^{(i)} - (\cdot)^{(i)} \right)^2 \right\}.$$

For the operator $\mathcal{T}_q^{(i)}$ in Eq. (8) we have

$$\mathcal{T}_q^{(i)} g_{x^{(-i)}}(\cdot) = \exp\left\{ -\frac{1}{2\sigma^2} \sum_{j=1}^m \left( x^{(j)} - (\cdot)^{(j)} \right)^2 \right\} \left( \frac{1}{\sigma^2}(x^{(i)} - (\cdot)^{(i)}) + (\log q_{t(x^{(-i)})})'(x^{(i)}) \right).$$

Thus, the operator $\mathcal{A}_t$ decomposes as

$$
\begin{aligned}
\frac{1}{m} \sum_{i=1}^m \mathcal{T}_q^{(i)} g_{x^{(-i)}}(\cdot) &= \exp\left\{ -\frac{1}{2\sigma^2} \sum_{j=1}^m \left( x^{(j)} - (\cdot)^{(j)} \right)^2 \right\} \\
&\quad \frac{1}{m} \sum_{i=1}^m \left\{ \frac{1}{\sigma^2} \left( x^{(i)} - (\cdot)^{(i)} \right) + (\log q_{t(x^{(-i)})})'(x^{(i)}) \right\}.
\end{aligned}
$$

For a proof of consistency, in our setting the Stein operators are only applied to the observations $z_1, \ldots, z_n$ and hence, for our applications, pointwise consistent estimation suffices, in the sense that that for $i = 1, \ldots, m$, $\widehat{s}_{t,N}^{(i)} = \widehat{s}_{t,N}^{(i)}(x^{(i)})$ is a consistent estimator of the univariate score function $s_t^{(i)} = \{\log(q(x^{(i)})|t(x^{(-i)}))\}'$. Score matching estimators often satisfy not only consistency but also asymptotic normality, see for example Song et al. [2020]. Such an assumption is required for Theorem 3.2; recall that we use the notation $\hat{s}_{t,N} = (\hat{s}_{t,N}(x^{(i)}), i \in [m])$. To prove Theorem 3.2 we re-state it for convenience.

**Theorem B.3.** *Assume that the score function estimator vector $\hat{s}_{t,N}$ is asymptotically normal with mean $0$ and covariance matrix $N^{-1}\Sigma_s$. Then $\text{NP-KSD}_t^2(G\|p)$ converges in probability to $\text{KSD}_t^2(q_t\|p)$ at rate at least $\min(B^{-\frac{1}{2}}, N^{-\frac{1}{2}})$.*

**Proof.** We have from Eq. (5)

$$\text{NP-KSD}_t^2(G\|p) = \mathbb{E}_{\boldsymbol{x},\tilde{\boldsymbol{x}}\sim p}[\langle \widehat{\mathcal{A}}_{t,N}^B k(\boldsymbol{x},\cdot), \widehat{\mathcal{A}}_{t,N}^B k(\tilde{\boldsymbol{x}},\cdot)\rangle_{\mathcal{H}}.$$

Expanding this expression, with $\mathcal{A}_t$ denoting the score Stein operator in Eq. 7 for the conditional distribution $q_t$,

$$
\begin{aligned}
\text{NP-KSD}_t^2(G\|p) &= \mathbb{E}_{\boldsymbol{x},\tilde{\boldsymbol{x}}\sim p}[\langle \mathcal{A}_t k(\boldsymbol{x},\cdot), \mathcal{A}_t k(\tilde{\boldsymbol{x}},\cdot)\rangle_{\mathcal{H}} \\
&\quad + \mathbb{E}_{\boldsymbol{x},\tilde{\boldsymbol{x}}\sim p}[\langle (\widehat{\mathcal{A}}_{t,N}^B - \mathcal{A}_t)k(\boldsymbol{x},\cdot), \mathcal{A}_t k(\tilde{\boldsymbol{x}},\cdot)\rangle_{\mathcal{H}} \\
&\quad + \mathbb{E}_{\boldsymbol{x},\tilde{\boldsymbol{x}}\sim p}[\langle \mathcal{A}_t k(\boldsymbol{x},\cdot), (\widehat{\mathcal{A}}_{t,N}^B - \mathcal{A}_t)k(\tilde{\boldsymbol{x}},\cdot)\rangle_{\mathcal{H}} \\
&\quad + \mathbb{E}_{\boldsymbol{x},\tilde{\boldsymbol{x}}\sim p}[\langle (\widehat{\mathcal{A}}_{t,N}^B - \mathcal{A}_t)k(\boldsymbol{x},\cdot), \widehat{(\mathcal{A}}_{t,N}^B - \mathcal{A}_t)k(\tilde{\boldsymbol{x}},\cdot)\rangle_{\mathcal{H}} \\
&= \text{KSD}^2(q_t\|p) + 2\,\mathbb{E}_{\boldsymbol{x},\tilde{\boldsymbol{x}}\sim p}[\langle (\widehat{\mathcal{A}}_{t,N}^B - \mathcal{A}_t)k(\boldsymbol{x},\cdot), \mathcal{A}_t k(\tilde{\boldsymbol{x}},\cdot)\rangle_{\mathcal{H}} \\
&\quad + \mathbb{E}_{\boldsymbol{x},\tilde{\boldsymbol{x}}\sim p}[\langle (\widehat{\mathcal{A}}_{t,N}^B - \mathcal{A}_t)k(\boldsymbol{x},\cdot), (\widehat{\mathcal{A}}_{t,N}^B - \mathcal{A}_t)k(\tilde{\boldsymbol{x}},\cdot)\rangle_{\mathcal{H}}
\end{aligned}
$$

where we used the symmetry of the inner product in the last step. Now, for any function $g$ for which the expression is defined,

$$(\widehat{\mathcal{A}}_{t,N}^B - \mathcal{A}_t)g(\boldsymbol{x}) = (\widehat{\mathcal{A}}_{t,N}^B - \widehat{\mathcal{A}}_{t,N})g(\boldsymbol{x}) + (\widehat{\mathcal{A}}_{t,N} - \mathcal{A}_t)g(\boldsymbol{x}) \tag{23}$$

recalling that $\widehat{\mathcal{A}}_{t,N}$ is the Stein operator using the estimated conditional score function $\hat{s}_{t,N}$ with the estimation based on $N$ synthetic observations.

To analyse Eq. 23 we first consider $(\widehat{\mathcal{A}}_{t,N} - \mathcal{A}_t)g(\boldsymbol{x})$;

$$\widehat{\mathcal{A}}_{t,N}g(\boldsymbol{x}) - \mathcal{A}_t g(\boldsymbol{x}) = \frac{1}{m}\sum_{i=1}^m g(x^{(i)})(\widehat{s}_{t,N}^{(i)}(x^{(i)}) - s_t^{(i)}(x^{(i)})). \tag{24}$$

As for $x^{(1)}, \ldots, x^{(m)}$ the vector $\hat{s}_{t,N} = (\hat{s}_{t,N}(x^{(i)}), i \in [m])$ is assumed to be asymptotically normal with mean $0$ and covariance matrix $N^{-1}\Sigma_s$, it follows from Eq. 24 that, asymptotically, $\sqrt{N}(\widehat{\mathcal{A}}_{t,N}g(\boldsymbol{x}) - \mathcal{A}_{q_t}g(\boldsymbol{x}))$ has a multivariate normal distribution and, in particular, $(\widehat{\mathcal{A}}_{t,N}g(\boldsymbol{x}) - \mathcal{A}_{q_t}g(\boldsymbol{x}))$ has fluctuations of the order $N^{-\frac{1}{2}}$.

For the term $(\widehat{\mathcal{A}}_{t,N}^B - \widehat{\mathcal{A}}_{t,N})g(\boldsymbol{x})$ in Eq. 23 we have

$$
\begin{aligned}
(\widehat{\mathcal{A}}_{t,N}^B - \widehat{\mathcal{A}}_{t,N})g(\boldsymbol{x}) &= \frac{1}{B}\sum_{b=1}^B \mathcal{T}_{t,N}^{(i_b)}g(x) - \mathcal{A}_{t,N}g(\boldsymbol{x}) \\
&= \sum_{i=1}^m \left\{ \frac{1}{B}\sum_{b=1}^B \mathcal{T}_{t,N}^{(i_b)}g(\boldsymbol{x})\mathbb{1}(i_b = i) - \frac{1}{m}\mathcal{T}_{t,N}^{(i)}g(\boldsymbol{x}) \right\}.
\end{aligned}
$$

Let $k_i = \sum_{b=1}^B \mathbb{1}(i_b = i)$ the number of times that $i$ is re-sampled. Then $\mathbb{E}(k_i) = \frac{B}{m}$ and we have

$$
\begin{aligned}
(\widehat{\mathcal{A}}_{t,N}^B - \widehat{\mathcal{A}}_{t,N})g(\boldsymbol{x}) &= \sum_{i=1}^m \mathcal{T}_{t,N}^{(i)}g(\boldsymbol{x})\left\{ \frac{1}{B}k_i - \frac{1}{m} \right\} \\
&= \frac{1}{B}\sum_{i=1}^m \mathcal{T}_{t,N}^{(i)}g(\boldsymbol{x})\left\{ k_i - \mathbb{E}(k_i) \right\}.
\end{aligned}
$$

This term is known to be approximately mean zero normal with finite variance $\Sigma(\hat{s}_{t,N}; g)$ (which depends on $\hat{s}_{t,N}$ and $g$) of order $B^{-1}$, see for example Holmes and Reinert [2004], where an explicit bound on the distance to normal is provided. This asymptotic normality holds for the operator given the estimated conditional score function. As the bootstrap samples are drawn independently of the score function estimator, without conditioning, the unconditional distribution is a mixture of normal distributions. For an estimator $\hat{s}_{t,N}$ which is asymptotically normally distributed, the variances $\Sigma(\hat{s}_{t,N}; g)$ will converge to $\Sigma(s_t; g)$.

Thus, with Eq. 23,

$$\mathbb{E}_{\boldsymbol{x}, \tilde{\boldsymbol{x}} \sim p}[\langle (\widehat{\mathcal{A}}_{t,N}^B - \mathcal{A}_t)k(\boldsymbol{x}, \cdot), \mathcal{A}_t k(\tilde{\boldsymbol{x}}, \cdot) \rangle_{\mathcal{H}}]$$
$$= \mathbb{E}_{\boldsymbol{x}, \tilde{\boldsymbol{x}} \sim p}[\langle (\widehat{\mathcal{A}}_{t,N}^B - \widehat{\mathcal{A}}_{t,N})k(\boldsymbol{x}, \cdot), \mathcal{A}_t k(\tilde{\boldsymbol{x}}, \cdot) \rangle_{\mathcal{H}}] + \mathbb{E}_{\boldsymbol{x}, \tilde{\boldsymbol{x}} \sim p}[\langle (\widehat{\mathcal{A}}_{t,N} - \mathcal{A}_t)k(\boldsymbol{x}, \cdot), \mathcal{A}_t k(\tilde{\boldsymbol{x}}, \cdot) \rangle_{\mathcal{H}}]$$

with the first term is approximately a variance mixture of mean zero normals tending to 0 in probability at rate at least $B^{-\frac{1}{2}}$ as $B \to \infty$, and the second term approximately a mean zero normal variable tending to 0 in probability at rate at least $N^{-\frac{1}{2}}$ as $N \to \infty$.

It remains to consider

$$\mathbb{E}_{\boldsymbol{x}, \tilde{\boldsymbol{x}} \sim p}[\langle (\widehat{\mathcal{A}}_{t,N}^B - \mathcal{A}_t)k(\boldsymbol{x}, \cdot), (\widehat{\mathcal{A}}_{t,N}^B - \mathcal{A}_t)k(\tilde{\boldsymbol{x}}, \cdot) \rangle_{\mathcal{H}}.$$

With Eq. 23 we have

$$\mathbb{E}_{\boldsymbol{x}, \tilde{\boldsymbol{x}} \sim p}[\langle (\widehat{\mathcal{A}}_{t,N}^B - \mathcal{A}_t)k(\boldsymbol{x}, \cdot), (\widehat{\mathcal{A}}_{t,N}^B - \mathcal{A}_t)k(\tilde{\boldsymbol{x}}, \cdot) \rangle_{\mathcal{H}}$$

$$= \mathbb{E}_{\boldsymbol{x}, \tilde{\boldsymbol{x}} \sim p}[\langle (\widehat{\mathcal{A}}_{t,N}^B - \widehat{\mathcal{A}}_{t,N})k(\boldsymbol{x}, \cdot), (\widehat{\mathcal{A}}_{t,N}^B - \widehat{\mathcal{A}}_{t,N})k(\tilde{\boldsymbol{x}}, \cdot) \rangle_{\mathcal{H}} \tag{25}$$

$$+ \mathbb{E}_{\boldsymbol{x}, \tilde{\boldsymbol{x}} \sim p}[\langle (\widehat{\mathcal{A}}_{t,N}^B - \widehat{\mathcal{A}}_{t,N})k(\boldsymbol{x}, \cdot), (\widehat{\mathcal{A}}_{t,N} - \mathcal{A}_t)k(\tilde{\boldsymbol{x}}, \cdot) \rangle_{\mathcal{H}} \tag{26}$$

$$+ \mathbb{E}_{\boldsymbol{x}, \tilde{\boldsymbol{x}} \sim p}[\langle (\widehat{\mathcal{A}}_{t,N} - \mathcal{A}_t)k(\boldsymbol{x}, \cdot), (\widehat{\mathcal{A}}_{t,N}^B - \widehat{\mathcal{A}}_{t,N})k(\tilde{\boldsymbol{x}}, \cdot) \rangle_{\mathcal{H}} \tag{27}$$

$$+ \mathbb{E}_{\boldsymbol{x}, \tilde{\boldsymbol{x}} \sim p}[\langle (\widehat{\mathcal{A}}_{t,N} - \mathcal{A}_t)k(\boldsymbol{x}, \cdot), (\widehat{\mathcal{A}}_{t,N} - \mathcal{A}_t)k(\tilde{\boldsymbol{x}}, \cdot) \rangle_{\mathcal{H}}. \tag{28}$$

In Xu and Reinert [2021], Proposition 2, the following result is shown, using the notation as above.

Let

$$Y = \frac{1}{B^2} \sum_{s,t \in [m]} (k_s k_t - \mathbb{E}(k_s k_t)) h_x(s, t).$$

Assume that $h_x$ is bounded and that $Var(Y)$ is non-zero. Then if $Z$ is mean zero normal with variance $Var(Y)$, there is an explicitly computable constant $C > 0$ such that for all three times continuously differentiable functions $g$ with bounded derivatives up to order 3,

$$|\mathbb{E}[g(Y)] - \mathbb{E}[g(Z)]| \leq \frac{C}{B}.$$

Moreover, using Equations (17)-(21) from Ouimet [2021], it is easy to see that $Var(Y)$ is of the order $B^{-1}$. Hence, Term (25) tends to 0 in probability at rate at least $B^{-1}$. Similarly, using that the bootstrap sampling is independent of the score function estimation, Terms (26) and (27) tend to 0 in probability at rate at least $(NB)^{-\frac{1}{2}}$. For Term (28), from Eq. (24),

$$\mathbb{E}_{\boldsymbol{x}, \tilde{\boldsymbol{x}} \sim p}[\langle (\widehat{\mathcal{A}}_{t,N} - \mathcal{A}_t)k(\boldsymbol{x}, \cdot), (\widehat{\mathcal{A}}_{t,N} - \mathcal{A}_t)k(\tilde{\boldsymbol{x}}, \cdot) \rangle_{\mathcal{H}}]$$

$$= \frac{1}{m^2} \sum_{i=1}^m \sum_{j=1}^m \mathbb{E}_{\boldsymbol{x}, \tilde{\boldsymbol{x}} \sim p}[\langle (\widehat{s}_{t,N}^{(i)}(x^{(i)}) - s_t^{(i)}(x^{(i)}))k(x^{(i)}, \cdot), (\widehat{s}_{t,N}^{(j)}(x^{(j)}) - s_t^{(j)}(x^{(j)}))k(x_j, \cdot) \rangle_{\mathcal{H}}].$$

If $\hat{s}_{t,N}$ is approximately normal as hypothesised, then the inner product is approximately a covariance of order $N^{-1}$, and hence the overall contribution from Term (28) is of order at most $N^{-1}$. This finishes the proof. $\square$

## C  Additional experimental details and results

### C.1  Additional experiments

**Runtime**  The computational runtime for all tests are shown in Table 3. **MMDAgg** runtime is also shown as a comparison. From the result, we can see that NP-KSD runs generally slower than the

| NP-KSD resampling size | B=5 | B=10 | B=20 | B=40 | (MMDAgg) |
|:---:|:---:|:---:|:---:|:---:|:---:|
| Runtime(s) | 4.65 | 6.56 | 8.43 | 10.44 | 5.02 |
| Rejection Rate | 0.24 | 0.40 | 0.51 | 0.55 | 0.27 |

Table 3: Computational runtime for various re-sample size B: observed sample size $n = 100$; bootstrap size $b = 200$; dimension $m = 40$. The rejection rate is used for power comparison; higher rejection rates indicate higher power.

permutation-based test **MMDAgg**. This is mainly due to the learning of conditional score functions and the Monte-Carlo based bootstrap procedure. As the re-sample size $B$ increases, NP-KSD test requires longer runtime. However, the rejection rate $B = 20$ is close to that of $B = 40$ (echoing the results shown in Figure 1(d)). While **MMDAgg** generally has faster computational runtime, it has lower test power, which is only comparable to that of NP-KSD with $B = 5$, at which the runtime advantage is not that obvious.

**Training on synthetic distributions**   We also train the deep generative models on the synthetic distributions studied in Section 4 and perform model assessment on the trained models. We consider the standard Gaussian and Mixture of two-component Gaussian problems. We train a generative adversarial network with multi-layer perceptron (**GAN_MLP**)[9] and a variational auto-encoder (VAE) [Kingma and Welling, 2013]. A Noise-Contrastive Score Network **NCSN** is also trained to learn the score function, followed by annealed Langevin dynamics [Song and Ermon, 2019, 2020]. All trainin is carried out via the optimiser Adam [Kingma and Ba, 2014] with adaptive learning rate. The rejection rates are reported in Table 4.

As shown in Table 4, the unimodal Gaussian distribution is easier to be learned by the generative modelling procedures, as compared to the two-component Mixture of Gaussian (MoG) model. As a result, the NP-KSD_m testing procedure shows a higher rejection rate on trained MoG generative models compared to that of Gaussian.[10]. However, as these deep models are not designed for training and sampling a simple low-dimensional distribution, it is not surprising that the procedure produce samples that not pass the NP-KSD tests.

Inspired by the settings in Gorham and Mackey [2017], where KSD is used to measure sample quality, we also apply NP-KSD tests on the Stochastic Gradient Langevin Dynamics (SGLD) [Welling and Teh, 2011] sampling procedure studied in Gorham and Mackey [2017]; in Gorham and Mackey [2017], SGLD is referred to as Stochastic Gradient Fisher Scoring (SGFS). SGLD is capable of sampling unimodal distributions, while it can have problems sampling multimodal data. The rejection rates shown in Table 4 are slightly higher than the test level for MoG, while the type-I error is well controlled for the Gaussian case. Generated samples from SGLD are visualised Figure 2, illustrating that the SGLD samples look plausible for the Gaussian model, but less so for the MoG model.

| | GAN_MLP | VAE | NCSN | SGLD | Real |
|:---:|:---:|:---:|:---:|:---:|:---:|
| Gaussian | 0.36 | 0.61 | 0.25 | 0.06 | 0.03 |
| MoG | 0.78 | 0.92 | 0.45 | 0.12 | 0.04 |

Table 4: NP-KSD_m rejection rate: observed sample size $n = 100$; bootstrap size is 200. Here a low rejection rate indicates a good type-1 error. Among deep generative models NCSN performs best on both tasks but it still has a very high rejection rate. SGLD outperforms the deep generative models.

---

[9]DCGAN studied in the main text is particularly useful for the (high-dimensional) image dataset due to the convolutional neural network (CNN) layers; DCGAN is not applicable for the problem in $\mathbb{R}^2$.

[10]We note that NP-KSD and NP-KSD_m with summary statistics taken to be the mean are equivalent in the two-dimensional problem.

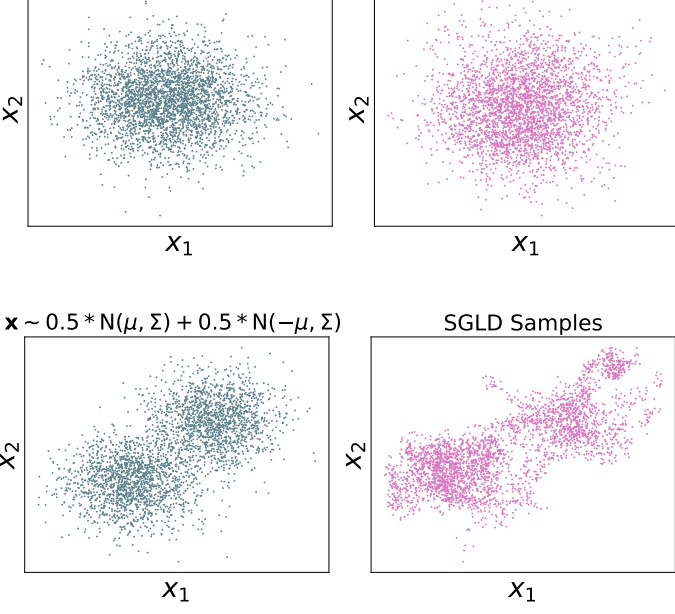

Figure 2: Visualisation of samples generated from Stochastic Gradient Langevin Dynamics (SGLD); top: Gaussian model, bottom: MoG model.

## C.2 Data visualisation

In Figure 3 and Figure 4 we show samples from the MNIST and CIFAR10 dataset, together with samples from trained generative models, respectively.

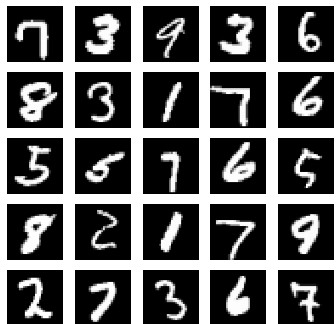

(a) Real MNIST samples

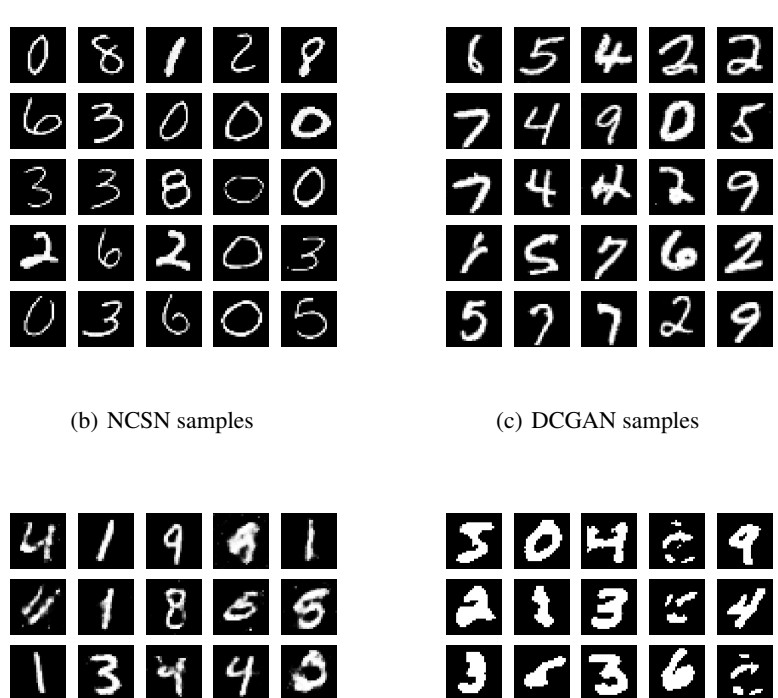

(b) NCSN samples

(c) DCGAN samples

(d) GAN samples

(e) VAE samples

Figure 3: MNIST samples

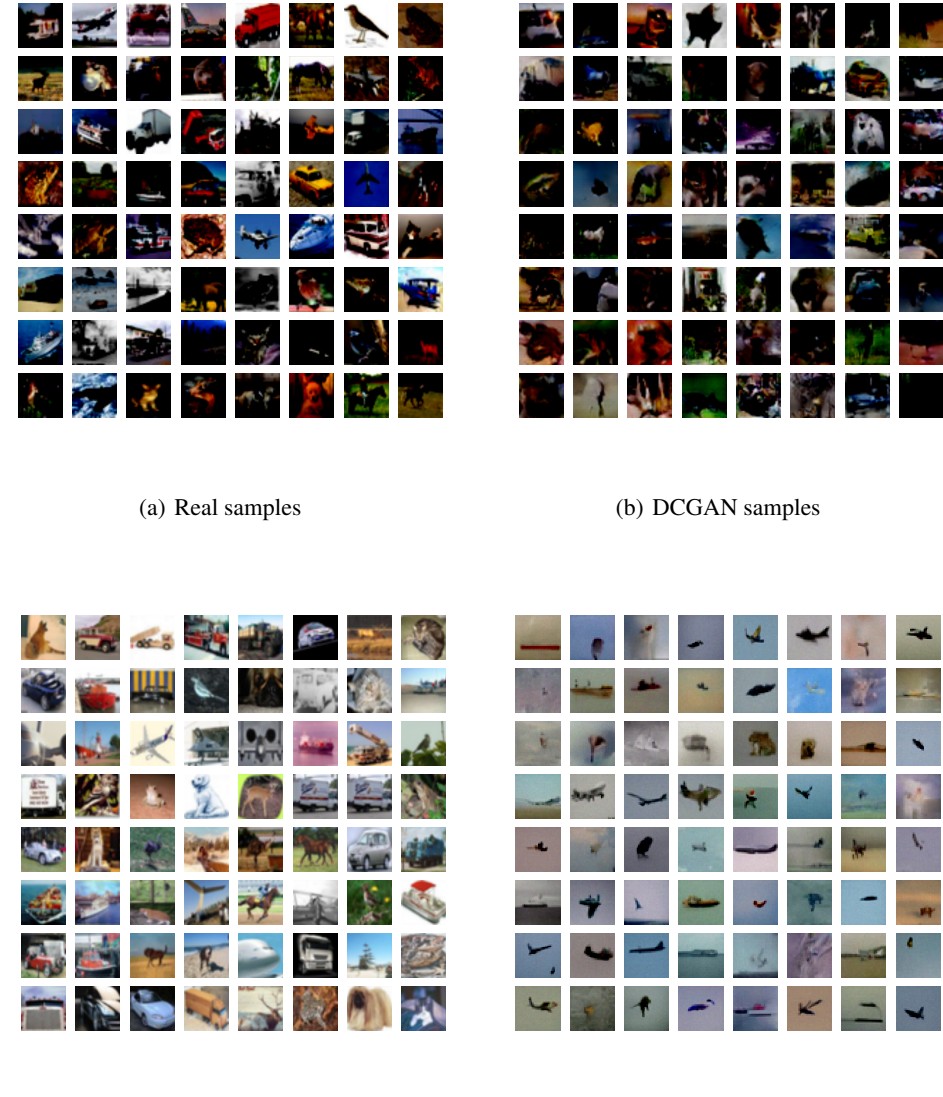

(a) Real samples

(b) DCGAN samples

(c) CIFAR10.1 samples

(d) NCSN samples

Figure 4: CIFAR10 samples

# D  Equivalence to the multivariate score-Stein operator

Here we show that the operator in Eq. 7 and the corresponding multivariate score-Stein operator in Eq. 2 are equivalent, when they exist; the difference being the factor $\frac{1}{m}$. First we recall the set-up for score-Stein operators. Let $q$ with smooth support $\Omega_q$ be differentiable. The score function of $q$ is the function

$$\mathbf{s}_q = \mathcal{T}_{\nabla,q}1 = \nabla \log q = \frac{\nabla q}{q}$$

(with the convention that $\mathbf{s}_q \equiv 0$ outside of $\Omega_q$). The score-Stein operator is the vector-valued operator

$$\mathcal{A}_q = \nabla + \mathbf{s}_q \mathbf{I}_m \tag{29}$$

acting on differentiable functions $g : \mathbb{R}^m \to \mathbb{R}$, with $\mathbf{I}_m$ denoting the $m \times m$ identity matrix.

**Proposition D.1.** *When they exist, then the operators in Eq. 7 and in Eq. 2 differ only by a factor $\frac{1}{m}$.*

**Proof.** Writing $\partial_i$ for the derivative in direction $x^{(i)}$, the score operator acting on differentiable functions $g : \mathbb{R}^m \to \mathbb{R}$ can be written as

$$\mathcal{A}_p g(x) = \sum_{i=1}^{m} \left\{ \partial_i g(x) + g(x) \partial_i (\log q(x)) \right\}. \tag{30}$$

Now, for $i \in [m]$,

$$q(x) = q(x^{(i)} | x^{(j)}, j \neq i) q(x^{(j)}, j \neq i)$$

and hence

$$\partial_i (\log q(x)) = \partial_i \log q(x^{(i)} | x^{(j)}, j \neq i).$$

The assertion follows. □

**Example D.2** (Bi-variate Gaussian). Consider $x = (x^{(1)}, x^{(2)})^\top \in \mathbb{R}^2$, i.e. $m = 2$ and $x \sim \mathcal{N}(\mu, \Sigma)$ where $\mu = (\mu^{(1)}, \mu^{(2)})^\top \in \mathbb{R}^2$, $\Sigma = \begin{pmatrix} 1 & \sigma \\ \sigma & 1 \end{pmatrix}$. With the corresponding precision matrix $\Sigma^{-1} = \frac{1}{1-\sigma^2} \begin{pmatrix} 1 & -\sigma \\ -\sigma & 1 \end{pmatrix}$, it is easy to check $Q^{(1)}(X^{(1)} | X^{(2)} = x^{(2)}) \sim \mathcal{N}(\mu^{(1)} + \sigma(x^{(2)} - \mu^{(2)}), 1 - \sigma^2)$.
For a bi-variate differentiable test function $g : \mathbb{R}^2 \to \mathbb{R}$, applying the Stein operator of the form in Section 3,

$$
\begin{aligned}
\mathcal{A}g(x^{(1)}, x^{(2)}) &= \frac{1}{2} \left\{ \mathcal{T}^{(1)} g_{x^{(2)}}(x^{(1)}) + \mathcal{T}^{(2)} g_{x^{(1)}}(x^{(2)}) \right\} \\
&= \frac{1}{2} \left\{ (g_{x^{(2)}})'(x^{(1)}) - \frac{x^{(1)} - \mu^{(1)} - \sigma(x^{(2)} - \mu^{(2)})}{1 - \sigma^2} g_{x^{(2)}}(x^{(1)}) \right. \\
&\quad \left. + (g_{x^{(1)}})'(x^{(2)}) - \frac{x^{(2)} - \mu^{(2)} - \sigma(x^{(1)} - \mu^{(1)})}{1 - \sigma^2} g_{x^{(1)}}(x^{(2)}) \right\} \\
&= \frac{1}{2} \left\{ \partial_1 g(x^{(1)}, x^{(2)}) - \frac{x^{(1)} - \mu^{(1)} - \sigma(x^{(2)} - \mu^{(2)})}{1 - \sigma^2} g_{x^{(2)}}(x^{(1)}) \right. \\
&\quad \left. + \partial_2 g(x^{(1)}, x^{(2)}) - \frac{x^{(2)} - \mu^{(2)} - \sigma(x^{(1)} - \mu^{(1)})}{1 - \sigma^2} g_{x^{(1)}}(x^{(2)}) \right\} \\
&= \frac{1}{2} \left\{ \nabla \times g(x^{(1)}, x^{(2)}) - \Sigma^{-1} (x^{(1)} - \mu^{(1)}, x^{(2)} - \mu^{(2)})^T g(x^{(1)}, x^{(2)}) \right\}
\end{aligned}
$$

where $\partial_i$ denotes the derivative with respect to $x^{(i)}$. Thus, we recover the score operator given in Eq. 2.

# E   Energy-based models and score matching

An Energy-based model (EBM) has density given by

$$q(x) = \frac{1}{Z} \exp\{-E(x)\}, \tag{31}$$

where $Z$ is the (generally) intractable normalisation constant (or partition function) and $E(x)$ is called the energy function. EBMs [LeCun et al., 2006] have been used in machine learning contexts for modelling and learning deep generative models. In particular, learning and training EBMs has been studied in machine learning, see for example Song and Kingma [2021]. One of the most popular and relatively stable training objectives is the score-matching (SM) objective given in Eq. (10) [Hyvärinen, 2005],

$$J(p\|q) = \mathbb{E}_p \left[ (\log p(x)' - \log q(x)')^2 \right].$$

For an EBM, the SM objective only requires computing $\nabla E(x)$ and $\nabla \cdot \nabla E(x)$ (or $\Delta E(x)$), but it does not require computing the partition function $Z$. Moreover, by learning the SM objective, we can obtain $\nabla \log q(x)$ directly, to construct an approximate Stein operator. We use this method for the univariate score function estimation.

# F   More on kernel-based hypothesis tests

## F.1   Maximum-mean-discrepancy tests

**Maximum-mean-discrepancy (MMD)**   has been introduced as a kernel-based method to tackle two-sample problems [Gretton et al., 2007], utilising the rich representation of the functions in a reproducing kernel Hilbert space (RKHS) via a kernel mean embedding. Let $k : \mathcal{X} \times \mathcal{X} \to \mathbb{R}$ be the kernel associated with RKHS $\mathcal{H}$. The kernel *mean embedding* of a distribution $p$ induced by $k$ is defined as the function

$$\mu_p := \mathbb{E}_{x \sim p}[k(x, \cdot)] \in \mathcal{H}, \tag{32}$$

whenever $\mu_p$ exist. The kernel mean embedding in Eq.32 can be estimated empirically from independent and identically distributed (i.i.d.) samples. Given $x_1, \ldots, x_n \sim p$:

$$\widehat{\mu}_p := \frac{1}{n} \sum_{i=1}^{n} k(x_i, \cdot) \tag{33}$$

replacing $p$ by its empirical counterpart $\widehat{p} = \frac{1}{n} \sum_{i=1}^{n} \delta_{x_i}$ where $\delta_{x_i}$ denotes the Dirac measure at $x_i \in \mathcal{X}$. For i.i.d. samples, the empirical mean embedding $\widehat{\mu}_p$ is a $\sqrt{n}$-consistent estimator for $\mu_p$ in RKHS norm [Tolstikhin et al., 2017], and with $n$ denoting the number of samples, $\|\mu_p - \widehat{\mu}_p\|_{\mathcal{H}} = O_p(n^{-\frac{1}{2}})$. When the sample size $n$ is small, the estimation error may not be negligible.

The MMD between two distributions $p$ and $q$ is defined as

$$\begin{aligned} \mathrm{MMD}(p\|q; \mathcal{H}) &= \sup_{\|f\|_{\mathcal{H}} \leq 1} \mathbb{E}_{x \sim p}[f(x)] - \mathbb{E}_{\tilde{x} \sim q}[f(\tilde{x})] \\ &= \sup_{\|f\|_{\mathcal{H}} \leq 1} \langle f, \mu_p - \mu_q \rangle_{\mathcal{H}} = \|\mu_p - \mu_q\|_{\mathcal{H}}. \end{aligned} \tag{34}$$

One desirable property for MMD is to be able to distinguish distributions in the sense that $\mathrm{MMD}(p\|q; \mathcal{H}) = 0 \iff p = q$[11]. This property can be achieved via *characteristic kernels* [Sriperumbudur et al., 2011].

It is often more convenient to work with the squared version of MMD:

$$\begin{aligned} \mathrm{MMD}^2(p\|q; \mathcal{H}) &= \|\mu_p - \mu_q\|_{\mathcal{H}}^2 = \langle \mu_p, \mu_p \rangle + \langle \mu_q, \mu_q \rangle - 2 \langle \mu_p, \mu_q \rangle \\ &= \mathbb{E}_{x, \tilde{x} \sim p} k(x, \tilde{x}) + \mathbb{E}_{y, \tilde{y} \sim q} k(y, \tilde{y}) - \mathbb{E}_{x \sim p, y \sim q} k(x, y). \end{aligned} \tag{35}$$

Given two sets of i.i.d. samples $\mathbb{S}_p = \{x_1, \ldots, x_n\} \overset{i.i.d.}{\sim} p$ and $\mathbb{S}_q = \{y_1, \ldots, y_l\} \overset{i.i.d.}{\sim} q$, an unbiased estimator of Eq.35, based on the empirical estimate of kernel mean embedding in Eq.33, is given by

$$\mathrm{MMD}_u^2(\mathbb{S}_p\|\mathbb{S}_q; \mathcal{H}) = \frac{1}{n(n-1)} \sum_{i \neq i'} k(x_i, x_{i'}) + \frac{1}{l(l-1)} \sum_{j \neq j'} k(y_j, y_{j'}) - \frac{2}{nl} \sum_{ij} k(x_i, y_j). \tag{36}$$

**A two-sample test (or two-sample problem)** aims to test the null hypothesis $H_0 : p = q$ against the alternative hypothesis $H_1 : p \neq q$. It has been shown that the *asymptotic* distribution of the $n$-scaled statistic $n \cdot \mathrm{MMD}_u^2(\mathbb{S}_p\|\mathbb{S}_q; \mathcal{H})$ under the null ($p = q$) is that of an infinite weighted sum of $\chi^2$-distribution [Gretton et al., 2012a, Theorem 12], while under the alternative ($p \neq q$), the $\sqrt{n}$-scaled statistic $\sqrt{n} \cdot \mathrm{MMD}_u^2(\mathbb{S}_p\|\mathbb{S}_q; \mathcal{H})$ is asymptotically normally distributed with the mean centered at $\mathrm{MMD}(p\|q; \mathcal{H}) > 0$. Thus, $n \cdot \mathrm{MMD}_u^2(\mathbb{S}_p\|\mathbb{S}_q; \mathcal{H})$ is taken as a test statistic to be compared against the *rejection threshold*. If the test statistic exceeds the rejection threshold, the empirical estimation of the MMD statistic is thought to exhibit significant departure from the null hypothesis so that $H_0$ is rejected.

As the null distribution is given by an infinite weighted sum of $\chi^2$ random variables which does not have a closed form expression, the null distribution is often simulated via a permutation procedure [Gretton et al., 2008]: Combine and order two sets of samples as $z_i = x_i, i \in [n]$ and $z_j = y_{j-n}, n+1 \leq j \leq n+l$. Let $\mu : [n+l] \to [n+l]$ be a permutation, and write $z^\mu = \{z_{\mu(1)}, \ldots z_{\mu(n+l)}\}$.

---

[11]Note that MMD is symmetric with respect to $p, q$, while KSD is not symmetric with respect to $p, q$.

| Sample size N | 20 | 50 | 100 | 200 | 500 | 1000 |
|---|---|---|---|---|---|---|
| MMD | 0.08 | 0.06 | **0.36** | **0.9** | **1.00** | **1.00** |
| MMDAgg | 0.06 | 0.07 | 0.02 | 0.03 | 0.02 | 0.05 |
| KSD | 0.07 | 0.04 | 0.04 | 0.02 | 0.08 | 0.06 |
| NP-KSD | 0.06 | 0.03 | 0.04 | 0.06 | 0.05 | 0.04 |

Table 5: Type-I error with increasing sample size $N$ for 100 tests. $H_0$ is Standard Gaussian with $m = 3$; $n = 50$; the test level is $\alpha = 0.05$; $n_{sim} = 500$. Bold values show uncontrolled type-I errors.

Then $z^\mu$ is re-split into $\mathbb{S}_p^\mu = \{z_i\}_{1 \leq i \leq n}$ and $\mathbb{S}_q^\mu = \{z_j\}_{n+1 \leq j \leq n+l}$. The permuted MMD is computed via Eq. 36 as

$$\mathrm{MMD}_u^2(z^\mu) = \mathrm{MMD}_u^2(\mathbb{S}_p^\mu \| \mathbb{S}_q^\mu; \mathcal{H}). \tag{37}$$

For $\mu_1, \ldots, \mu_B$, we obtain $\mathrm{MMD}_u^2(z^{\mu_1}), \ldots, \mathrm{MMD}_u^2(z^{\mu_B})$ and use these values to compute the empirical quantile of the test statistics $\mathrm{MMD}_u^2(\mathbb{S}_p \| \mathbb{S}_q; \mathcal{H})$.

To test whether the implicit generative model can generate samples following the same distribution as the observed sample, it is natural to consider the two-sample problem described above, which tests whether two sets of samples are from the same distribution. In the model assessment context, one set of samples (of size $N$) are generated from the implicit model, while the other set of samples (of size $n$) are observed.

The MMD test often assumes that the sample sizes $n$ and $l$ are equal; its asymptotic theoretical guarantees including consistency are valid under the regime that $n, l \to \infty$ [Gretton et al., 2009, 2012b, Jitkrittum et al., 2016]; also the relative model comparisons in Jitkrittum et al. [2018] only considered the cases $n = l$. In our setting, the sample size $l$ is usually denoted by $N$. For our model assessment problem setting, when $n$ is fixed and $N \to \infty$ is allowed to be asymptotically large, i.e. $n \ll N$, we find that the type-I error may not be controlled. Hence it is not always the case that MMD is able to pick up the distributional difference between two sets of samples under the null hypothesis. A simple experiment in Table 5 shows an example in which the type-I error is not controlled when $N$ is increasing. Hence, MMD is *not* used as comparison for NP-KSD.

The high type-I error of the MMD statistic as $N$ increases, shown in Table 5 can be heuristically explained as follows. Let $\{x_1, \ldots, x_n\}, \{\tilde{x}_1, \ldots, \tilde{x}_N\} \overset{i.i.d.}{\sim} p$ where the two sets of samples are generated from the same distribution. Let $\widehat{\mu}_{p,n} = \frac{1}{n} \sum_{i \in [n]} k(x_i, \cdot)$, and $\widehat{\mu}_{p,N} = \frac{1}{N} \sum_{j \in [N]} k(\tilde{x}_j, \cdot)$. The empirical MMD between $\widehat{\mu}_{p,n}$ and $\widehat{\mu}_{p,N}$ can be seen as

$$\|\widehat{\mu}_{p,n} - \widehat{\mu}_{p,N}\|_{\mathcal{H}}^2 = \|(\widehat{\mu}_{p,n} - \mu_p) - (\widehat{\mu}_{p,N} - \mu_p)\|_{\mathcal{H}}^2, \tag{38}$$

where MMD aims to detect the asymptotic equality of $(\widehat{\mu}_{p,n} - \mu_p)$ and $(\widehat{\mu}_{p,N} - \mu_p)$. When $n$ is small and fixed, and $n \ll N$, the difference is non-trivial and a rich-enough kernel is able detect this difference, leading to MMD rejecting the null hypothesis although it is true.

**MMDAgg, a non-asymptotic MMD-based test** Recently, Schrab et al. [2021] proposed an aggregated MMD test that can incorporate the setting $n \neq N$ as long as there exists a constant $C > 0$ such that $n \leq N \leq cn$. Under this condition, MMDAgg is a consistent non-asymptotic test with controlled type-I error, see Table 5. We use it as the competitor method in the main text and in Table 5; in Table 5 KSD is also included, as preferred method when the underlying null distribution is known.

The MMDAgg test statistic is computed by aggregating a set of MMD statistic based on different choices of kernel, e.g. Gaussian kernels with different bandwidth. MMDAgg takes into account a number of choices of bandwidth $\lambda \in \Lambda$ where $\Lambda$ is a finite set. Let $\widehat{M}_\lambda$ denote the empirical MMD using a kernel with bandwidth $\lambda$. Each $\lambda$ is weighted, via $w_\lambda$, where $\sum_{\lambda \in \Lambda} w_\lambda = 1$. In Schrab et al. [2021] (as well as in our implemented experiments), uniform weights are chosen; $w_\lambda \equiv w = \frac{1}{|\Lambda|}$. Denote by $B_1$ the number of samples used to simulate the null distribution for quantile estimation [12] and denote by $B_2$ the number of simulated samples used to estimate the empirical rejection probability.

---

[12]This is the same as notion $B$ in the main text as well as in Eq. 37.

Define $\hat{q}_{\lambda,1-\alpha}^{B_1}(z^{B_1})$ as the conditional empirical $(1-\alpha)$-quantile when MMD uses a kernel with bandwidth $\lambda$, estimated from the permutation procedure with $B_1$ permutations using Eq.37. Then for a fixed test level $\alpha$, a quantity $u$ is estimated via the bi-section method such that

$$\mathbb{P}\left(\max_{\lambda \in \Lambda}(\widehat{M}_\lambda - \hat{q}_{\lambda,uw}^{B_1}(z^{B_1})) > 0\right) \le \alpha. \tag{39}$$

We reject $H_0$ if for any $\lambda \in \Lambda$ and with the estimated $\hat{u}$, we have that $\widehat{M}_\lambda$ exceeds the rejection probability in Eq. 39; otherwise we do not reject $H_0$. In this way, MMDAgg does not only achieve the desired non-asymptotic type-I error but is also able to explore a wide range of kernels in order to produce stronger test power. The estimation and adjustment in Eq. 39 is absent in standard MMD testing procedures Gretton et al. [2012a] and is crucial for the MMDAgg being a non-asymptotic test.

### F.2 Wild-bootstrap on KSD testing procedures

In Chwialkowski et al. [2014] a wild bootstrap procedure is proposed to simulate the null distribution via so-called wild-bootstrap samples. For KSD, Chwialkowski et al. [2014] have shown weak asymptotic convergence to the null distribution for deterministic and bounded kernels. For the NP-KSD test statistic, wild-bootstrap samples do not necessarily converge to the null distribution, due to the estimation difference $(\hat{s}^t - s)$, creating a random Stein kernel for NP-KSD. Perhaps therefore unsurprisingly, the wild bootstrap procedure NP-KSD does not control the type-I error correctly. Instead, we consider a Monte Carlo procedure to simulate the null distribution of NP-KSD. While Monte Carlo estimation is more computationally intensive than wild-bootstrap, it is an accurate method by design.

Figure 5 illustrates this point. Figure 5(a) shows samples from a Gaussian distribution. The true density is plotted in red. Two score matching density estimates, SM1 and SM2, are calculated; SM1 presents a good fit whereas SM2 is a less accurate estimate. For KSD, which is applicable when the underlying null distribution is known, Figure 5(b) shows that the Monte Carlo distribution and the wild-bootstrap distribution are close and reach the same conclusion for the KSD test statistic. Using the well-fitting SM1 score density estimate, Figure 5(c) gives the Monte Carlo distribution and the wild-bootstrap distribution. The wild-bootstrap distribution is close to the wild-bootstrap distribution for the KSD. In contrast, it differs considerably from the Monte Carlo distribution and would reject the null hypothesis although it is true. Figure 5(d) shows a similar result for the not so well fitting estimate SM2. The wild-bootstrap distribution is now more spread out but the observed test statistic is still in the tail of this distribution, whereas it is close to the center of the Monte Carlo distribution. In the synthetic experiments for the MoG model in the main text, the model misspecification **NP-KSD_G** falls under this setting. These plots illustrate that using wild-bootstrap samples in this scenario could lead to erroneous conclusions. Hence we use Monte Carlo samples.

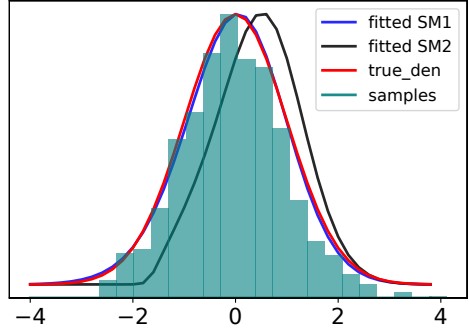

(a) Samples and fitted densities

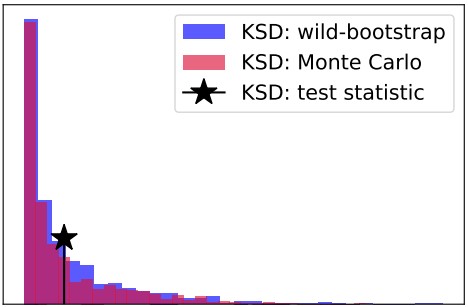

(b) Simulated null distributions from KSD

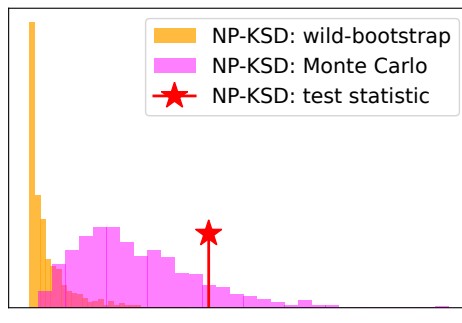

(c) Simulated null distributions from fitted SM1

(d) Simulated null distributions from fit SM2

Figure 5: Visualisation for NP-KSD and KSD testing procedures. For KSD, the wild-bootstrap distribution roughly agrees with the Monte Carlo distribution, whereas for NP-KSD, the wild-bootstrap distribution deviates strongly from the Monte Carlo distribution, indicating a danger of reaching an erroneous conclusion when using wild-bootstrap samples in this scenario.