# OpenReview forum: "A Kernelised Stein Statistic for Assessing Implicit Generative Models"
_NeurIPS.cc/2022/Conference — NeurIPS 2022 Accept_

### Official Review · Reviewer_MBab · 2022-07-06

**Rating:** 6
**Confidence:** 4
**Soundness:** 3 good
**Presentation:** 3 good
**Contribution:** 2 fair

**Summary:**

This paper proposes an adaptation of kernelized Stein discrepancy, called non-parametric KSD (NP-KSD), for assessing the implicit generative models. In particular, the author considered a scenario that (1) we do not have access to the (unnormalized) densities of the generative model; (2) the data dimension is large; (3) fixed number of true observations. To enable the assessment of this type of model, the author proposed a non-parametric Stein operator based (1) conditional distributions with summary statistics; (2) resampling the conditional dimensions; (3) score matching for gradient estimation. Together with the kernel trick and Monte Carlo based goodness-of-fit test, it results in the NP-KSD method.



**Questions:**

The questions are mainly the summary of what I have described in Weakness.

**Major**:
1. The motivation for using the goodness-of-fit test should be clearer.  E.g. advantages compared to two-sample test? Also, the implicit model has two application scenarios: (1) sampler, where we want to generate samples from a target distribution and (2) data synthesis, where we want to generate observations that mimic the real data samples, like GAN. From the context of this paper, it seems that it focuses on (2) instead of (1). It would be better to clarify this.
2. Do we have theoretical guarantees when NP-KSD =0 $\Rightarrow$ p=q? Does it only hold for equivalence class? What is the impact of equivalence class on the evaluation of the implicit model?
3. The concern about the novelty, as detailed in the weakness section. Particularly, if I understood correctly, the main framework of the proposed method is based on goodness-of-fit test but with estimated gradients from score matching. Those ideas are not new. So maybe consider elaborating more on the novel contributon?
4. For synthetic distribution experiments, why the proposed method is better than KSD? From my understanding, NP-KSD is an approximation of KSD.

**Minor**:
1. $\mathbb{R}^m$ in line 86 should be $\Omega_q$?
2. In the related work section, maybe consider adding citations regarding various types of Stein discrepancies and their usage in generative models.
3. I wonder why the proposed method is better than two-sample tests. Since the gradient estimation used in NP-KSD is based on the generated samples, it means NP-KSD does not have access to more information compared to the two-sample test. Where does this performance increase come from?

**Limitations:**

The author mentioned NP-KSD can distinguish distributions in equivalence classes. But the potential impact of the equivalence class to model evaluation can be discussed in more detail.  It would be great to consider some simple cases with simple summary statics like mean, and show some properties of the equivalence class.

**Strengths And Weaknesses:**

**Strength**:

The paper is clearly written and easy to follow. Although the motivation of the proposed method can be improved.
This paper aims to tackle an important problem: the evaluation of implicit models, which can have significant impacts to the current deep generative models. To sidestep the computational challenges, the author also provides several tricks to enable the NP-KSD. Theoretically, the author shows the consistency and convergence results of NP-KSD.

**Weakness**:

First, the motivation of the proposed method can be improved. For example, two-sample test is a typical method for evaluating implicit models. Why do we need goodness-of-fit test instead? Goodness-of-fit test is typically used when we have access to the density function, that is where its name comes from. Why do we want to use this for the implicit model with the gradient estimation? To me, it seems that we achieve what the two-sample test is designed for but with an alternative method. So a better motivation for goodness-of-fit test is needed.

There are several tricks used in deriving NP-KSD. The author has shown that NP-KSD satisfied Stein identity. However, a more interesting (or more important) aspect is to show when NP-KSD =0, it means p=q. Currently, I don't think it is true, especially with summary statistics and score matching. If not, then the goodness-of-fit test procedure will not be valid. The author mentioned equivalence class, but it is unclear how this will impact the evaluation of the implicit model. E.g. with a summary statistics, the distribution inside an equivalence class has similar visual quality?

Another concern is about its novelty. To me, the Stein operator \Tau is defined similarly as the Langevin-Stein operator but with conditional distributions. This is equivalent to the original Langevin-Stein discrepancy. This idea has also been explored in paper [1] and the author should consider citing it. If so, the main novelty of NP-KSD is score-matching, resampling and summary statistics. However, they seem to be more like tricks rather than novel methodologies.  Although I agree the theoretical consistency and convergence results give some guarantees of using the resampling and score matching, which seems to be novel.

[1] Singhal, R., Han, X., Lahlou, S., & Ranganath, R. (2019). Kernelized complete conditional Stein discrepancy. arXiv preprint arXiv:1904.04478.

---

> ### Author Response · Authors · 2022-07-31
> **Goodness-of-fit model assessment, Stein characterisation and theoretical justification**
>
>  Thank you for your review. We are pleased that you appreciate that we tackle an important problem. However what we propose is more than several tricks; it is a fundamentally different view to obtaining Stein operators and results in novel procedures.
>
>  Addressing your major concerns, indeed the paper focuses on GAN-type data generators, as indicated by the first three words in the abstract --- synthetic data generators are to be assessed. In standard statistical problems, usually the distribution underlying the observed data is the target; not so here.  Here our novel viewpoint comes into play: the ``target'' distribution is now the distribution from which the data generator generates samples. The test problem is to assess whether the observed sample could be viewed as coming from this target distribution.
>  This viewpoint is explained in lines 52-56.
>
>  Regarding theoretical guarantees, due to estimation error, it is not the case that NP-KSD=0 even when p=q. However, Proposition 3.1 and Theorem 3.2 give consistency guarantees. If p=q then KSD=0, and consequently, for $N$, the number of generated samples (and resamples $B$) tending to infinity, NP-KSD would approach 0 in probability, under the conditions stated in the theoretical results, as long as the score function is estimated consistently.
>
>  When we use the conditional score function, then the Stein operator will characterise the conditional distribution, now playing the role of p. Hence we can only hope for NP-KSD tending to 0 when p and q have the same conditional distributions given the summary statistic on which we condition. Of course, one can easily construct two distributions which are different but have the same conditional distribution (for example, a Poisson distribution conditioned on being at most 1 is a Bernoulli distribution, but in general Poisson and Bernoulli distributions are different). The effect of conditioning is illustrated for example in Figure 1 where NP-KSD and NP-KSD\_mean are compared: the first is unconditional, the second conditions on the mean. [You request such a comparison in your paragraph on limitations; it was already there.]
>
>  We note that as there is no parametric model available for the distribution which generates the data, KSD cannot be used, and this has been a main motivation for the development of NP-KSD. A key novelty is that we estimate the KSD for the generator $G$, which can be carried out to arbitrary precision as we can generate as many samples from $G$ as desired; Proposition 3.1 gives the theoretical justification and Theorem 3.2 extends it to the sampling scenario. Naively estimating this score function would be very difficult in high dimensions. Instead, we estimate univariate conditional score functions and combine them in Equation (7) to yield a Stein operator, which is the basis of the KSD test. When it is possible to sample from the conditional distributions (which we do not assume to be the case), a similar idea has been carried out in Singhal, R. et al. (2019) as you kindly indicated.
> Not citing this paper was an oversight which has now been amended. However, the idea of representing the Stein operator as a sum of Stein operators actually goes back to Reinert (2005), a paper which is cited in the references.
>
> To answer your question whether
> $\mathbb R^m$ in line 86 should be $\Omega_q$, the answer is that it is intended as stated;
>  as $\Omega_q \in \mathbb R^m $, the function is well defined. The requirement of the function belonging to the canonical Stein class ensures that the Stein identity holds.
>
>  We would like to emphasise that NP-KSD is developed for a very unbalanced situation in which standard two sample tests such as MMD tests and permutation tests fail. NP-KSD can treat the situation that only a small number $n$ of observations are available (say, one image, so that $n=1$),  whereas we can generate as many samples $N$ as desired, using the synthetic data generator. Typically $n \ll N$; the asymptotic results hold in the regime that $N$ tends to infinity, with $n$ fixed.
>
> The NP-KSD approach required novel theoretical underpinnings as well as thoughtful construction of test statistics. Moreover, it has a novel viewpoint, viewing as target distribution the unknown distribution which underlies the synthetic data generator, rather than the distribution from which the observed sample comes. Creating an empirical Stein operator and assessing its properties is also a novel addition to the literature which in our view goes far beyond ``tricks''. In summary, NP-KSD is able to solve an important problem using novel ideas and theoretical justifications.
>
> We hope that these explanations have clarified the contents of the paper and have addressed your concerns.

---

> > ### Comment · Reviewer_MBab · 2022-08-05
> > **Reply to the author**
> >
> > I am very appreciated for the detailed response from the author. It address some of my concerns. I still want to ask the following:
> >
> > 1. In the limitation, I was saying more like a detailed analysis on simple distribution with simple summary statistics. Your Poission example is a good example.
> >
> > 2. If for each of the conditional distribution and a Langevin-Stein operator is applied, the resulting Stein operator is equivalent to applying Langevin-Stein operator to the joint distribution, no? If so, this alone is not a novel Stein operator.

---

> > > ### Author Response · Authors · 2022-08-05
> > > **Further clarifications on the role of conditional distribution with summary statistics**
> > >
> > > Thank you for your question. As your question almost implies, the answer lies in the conditioning on the summary statistic $t$.  Without conditioning on a summary statistic, our Stein operator is the same as the Langevin Stein operator (up to scaling), which we show in Proposition D.1 in Appendix D.
> > >
> > > Then observing that we can decompose the Langevin Stein operator into operators which characterise univariate conditional distributions, we estimate the score function of the univariate conditional distributions, which is much easier than estimating the multivariate score function (which is what one would naturally do for the Langevin operator) in particular when the data are in high dimensions.
> > >
> > > The main use of this decomposition however and the key novelty of the paper lies in the conditioning on the summary statistic $t$. Each individual Stein operator now characterises a different conditional distribution. Looking at Eq.(29) in Appendix D, we use
> > >
> > > $$\partial_i \log (q ( x^{(i)} | t(x^{(-i)})) $$
> > > whereas if in the Langevin operator we condition on $t(x)$ we would obtain
> > >
> > > $$
> > > \partial_i \log (q (x | t(x))) =
> > >  \partial_i \{ \log (q (x^{(i)} | t(x), x^{(j)}, j \ne i )) q( x^{(j)}, j \ne i | t(x)) \}
> > > $$
> > > $$ =  \partial_i \log( q (x^{(i)} |x^{(j)}, j \ne i, t(x))  + \partial_i \log (q( x^{(j)}, j \ne i | t(x)) .
> > >  $$
> > >
> > >
> > > Thus the sum of the component-wise conditional Stein operators given $t(x^{(-i)})$ is not the same as the conditional Langevin operator given $t(x)$.
> > >
> > >
> > > To better understand the effect of the choice of summary statistic, it may be good to first not consider the summary statistic but just the conditional Stein operator. If we were to write our Stein operator as second-order operator then it is the generator of a Markov process which picks an index $I$ from $\{1, ..., m\}$ at random, and if $I=i$, replaces the observation $x^{(i)}$ by an observation $x^{(i)'}$ which is drawn from the conditional distribution of $x^{(i)}$, given $x^{(j)}, j \ne i$. This procedure is described for example in Reinert (2005). Our conditional Stein operator again picks an index $I$ from $\{1, ..., m\}$ at random, and if $I=i$, replaces the observation $x^{(i)}$ by an observation $x^{(i)'}$ which is now drawn from the conditional distribution of $x^{(i)}$, given $t(x^{(j)}, j \ne i)$. In general this is no longer the generator of a Markov process, but we show in the paper that we can still give theoretical guarantees for its behaviour, and it is useful as ingredient for our NP-KSD test statistic.
> > >
> > > Regarding the limitation phrasing, we appreciate that a better understanding of the choice of summary statistic would be very useful. In Section 5 of the paper we already discussed that the choice of summary statistic may have a large effect. We have now added a sentence, to read
> > >
> > >
> > > ``Future work will devote more attention on analysing the choice of summary statistic.''
> > >
> > >
> > > We hope that this explanation and addition alleviates your concerns.

---

> > > > ### Comment · Reviewer_MBab · 2022-08-08
> > > > **Reply to author's response**
> > > >
> > > > I appreciate the detailed response from the author. It addresses most of my concerns. I will raise my rating.

---

> > > > > ### Author Response · Authors · 2022-08-08
> > > > > **Reply**
> > > > >
> > > > > Thank you very much for the update. We are very pleased that we have addressed most of your concerns and you now support accepting it!

---

### Official Review · Reviewer_qNmx · 2022-07-11

**Rating:** 7
**Confidence:** 4
**Soundness:** 2 fair
**Presentation:** 4 excellent
**Contribution:** 3 good

**Summary:**

The paper presents a kernel Stein discrepancy based test for black box generative models.
Since the normal KSD cannot be applied, as the true score function of the black box model is unknown, the authors proposed a KSD variant where this unknown score function is estimated from samples from the model.
The resulting hypothesis test then is whether a given dataset that was used to fit the generative model is distributed according to the estimated score function of that model.
The score estimation is achieved via estimating a component wise distribution conditioned on all other components, through of summary statistics.
Some consistency results are provided.
An experimental evaluation on toy data and and simple real datasets shows some benefits against the original KSD test where applicable, and one MMD based baseline otherwise.

**Questions:**

A high level question is: Does it make sense to asses the quality of generative models via a hypothesis test? What do we really learn if we e.g. fail to reject the null hypothesis? What do we learn if we can reject it? It shouldn't actually be a surprise that it is easy for most statistical tests to reject the null. Some of the references I put above go further and explain what features in the data lead to the rejection. Something like this could be used to improve a generative model, but that is actually not done in this paper.


**Limitations:**



**Strengths And Weaknesses:**

A well written paper that addresses an important issue.
The proposed methodology is based around a few nice tricks on estimating the conditional distributions and carried our in a thorough manner.
In the current form of the paper, however, it is not clear whether the proposed methodology is really as useful as the authors claim. The experimental evaluation is not thorough, and lacks baselines and comparisons to alternative strategies.

A few concrete points:
* There is no like to like comparison with an MMD based method. Only MMDAgg as a very specific case; the point of MMDAgg is that it can select a kernel without having to sacrifice samples, but in the case of a generative model, we can generate many samples so it doesn't seem to be a well fitting baseline). What about e.g. a plain MMD test with the median heuristic, or with a learnt optimal kernel? What about e.g. the linear time tests that learn feature locations (E.g. https://arxiv.org/abs/1605.06796).
* It is actually not clear what is gained compared to an MMD based test. This surely depends on the number of samples drawn from the generator, as well as computational costs. See also question below. An empirical exploration of the performance of the various approaches as a function of drawn samples and compute would be very helpful. The authors only make some vague comments on this.
* It is not clear why we could not just use score matching to learn the score using a powerful neural network instead of the conditional distribution approach? This would allow using many samples from the generator as well.
* There are no empirical explorations of how the proposed score estimation approach scales in any direction. As that is crucial for this method to work, and since it is also a kind of harsh approximation, that weakens the paper. Related: what is the impact of the NN architecture for the score estimation in this case.

Missing paper references that are highly relevant, as the problem of assessing performance of generative models was first discussed here (among others)
* Statistical Model Criticism using Kernel Two Sample Tests by LLoyd et al
* Generative Models and Model Criticism via Optimized Maximum Mean Discrepancy by Sutherland et al

Minor:
* l97 The KSD was originally proposed in Chwialkowski et al 2016 and Liu at al 2016, not Gorham Mackey 2017
* l109 The wild bootstrap is only really used if there is correlation to break (e.g. for time series data). Otherwise plain permutation test is sufficient and has no free parameters
* l198 the normality result of Song is in a different context. Could you at least empirically explore this?

---

> ### Author Response · Authors · 2022-08-01
> **Model assessment via hypothesis testing, cocerns on score matching and MMD**
>
> You ask why we do not use MMD. MMD tests can be very useful in a two-sample problem for two sets of samples of comparable size, but in our setting the sample sizes can be very different. The sample size $n$ of the observed data is fixed and can be small, whereas the size $N$ of the sample generated by the synthetic data generator can be as large as desired. Prop.3.1 and Th.3.2, justifying NP-KSD, hold when N tends to infinity with n fixed. Our setting $n \ll N$   makes MMD  unsuitable as a comparison method.  The same argument applies to the optimised MMD test [1], the linear time version from Jitkrittum et al. (2016), and your suggested references [1] and [2]; for completeness, these are included in the new version. Lines 232-234 allude to issues arising in MMD when the sample size is small. Line 249, pointing to why MMD is not compared against, is now expanded:
>
> ``We note that MMD tests do not have controlled type-I error when $n\ll N$, thus are not suitable in this setting. However, MMD-based methods to compare and criticise two generative models have been explored [Lloyd and Gharahramani, 2015, Sutherland et al. 2017]. Detailed discussions and illustrations why MMD tests are not included in the comparison list are found in Appendix F.1.''
>
> Appendix F.1 also shows that MMD is not consistent in this highly imbalanced situation; Table 5 shows that its type-1 error does not reach the correct level.
> Instead, we compare to MMDAgg because it is a non-asymptotic test with controlled type-1 error even when $n \ll N$, see Table 5 in Appendix F (not because of its kernel selection feature without data splitting). Fig.1b assesses the effect of the number N of generated samples (on the x-axis). As discussed in lines 260-265, the power of NP-KSD increases faster with sample size than that of MMDAgg.
>
> Regarding learning the score using a neural network instead of the conditional distribution: we use the conditional scores because they are fast to estimate and lend themselves to a theoretical analysis: For sliced score matching, including one-dimensional score matching, Song et al. (2020) give conditions for the assumptions of  Prop.3.1 and Th.3.2 to hold. To assess the effect of score matching estimators, we both learned the score function directly, and the conditional score with the mean as the summary statistic, using NP-KSD and NP-KSD\_mean, see Fig. 1, Table 1 and 2. NP-KSD learns the full distribution with the score function parameterised by a particular deep neural network which corresponds to the conditional marginal distributions, as
> $$ \frac{\partial}{\partial x^{(i)}}\log q(x) = \frac{\partial}{\partial x^{(i)}}\log q(x^{(i)}, x^{(-i)}) =  \frac{\partial}{\partial x^{(i)}}\log q(x^{(i)}|x^{(-i)}) + \underset{=0}{\underbrace{\frac{\partial}{\partial x^{(i)}}q(x^{(-i)}) }} . $$
> Deriving further alternative score estimators and their theoretical behaviour, and assessing their performance in NP-KSD will be part of future work.
>
> Concerning the minor issues which you raised, we have amended the KSD reference in the new version; thank you for pointing this out. While the wild bootstrap process in Chwialkowski et al. (2014) can deal with non-independent data, we used it because it is part of the standard KSD procedure in Chwialkowski et al. (2016). However, estimating the scores violates the assumptions of the wild bootstrap; we show in Appendix F.2 that it can lead to erroneous results. A simple permutation test cannot be applied if one sample set is obtained.
>
> Finally, you raise the issue of the general value of a goodness-of-fit test. Comparing different generative models can be carried out via measuring sampling quality without significance level, e.g. in the setting of [1], or via a relative testing procedure, e.g. [3]. In contrast when only one particular generative model is of interest, then a goodness-of-fit testing procedure can be very useful as the model assessment can be performed just by comparing the p-value with the significance level.
> If the test does not reject a particular generator, then the test can aid the selection of reliable sample batches based on p-values, for example when small batches of high-quality samples are required, again without reference to any external models.
> Through inspecting accepted and rejected, guidance can be obtained for the development of alternative synthetic data generators. Exploring this further is part of our future work.
>
> We hope that these explanations have addressed your concerns.
>
> Additional references:
>
> [1] Sutherland, D. J. et al. (2017) Generative Models and Model Criticism via Optimized Maximum Mean Discrepancy. In ICLR (Poster).
>
> [2] Lloyd, J. R., and Ghahramani, Z. (2015) Statistical model criticism using kernel two-sample tests. Advances in Neural Information Processing Systems 28.
>
> [3] Kanagawa, H. et al.  (2019). A kernel Stein test for comparing latent variable models. arXiv preprint arXiv:1907.00586.

---

> > ### Comment · Reviewer_qNmx · 2022-08-03
> > **Update after author response**
> >
> > Many thanks to the authors for their detailed reply. I appreciate the points on  and that the MMD based methods I pointed out are unsuitable for the given context. This in fact addressed my most major concern re the usefulness in practice. The only caveat I have here is the question what would happen if we only sample  samples from the generator? Is the performance of such a test so much worse? A simple example to illustrate that would be great in my opinion. Also thank you for your thoughts on using goodness-of-fit tests to evaluate black box generative models in meaningful ways, I agree with most. I have increase my initial score.

---

> > > ### Author Response · Authors · 2022-08-04
> > > **An illustrative example**
> > >
> > >
> > > Many thanks for your update! Thank you also for your follow-up question. To clarify the disadvantage of using just samples from the generator, as a simple example we take the mean value as test statistics. Instead of an NP-KSD test, we now use a simple Monte Carlo test, with the MNIST data as an example. We calculate the mean value over each sample image (overall 28x28=784 pixel values), and we compare the mean of generated sample images with that of the real sample images.
> > > For each generator, we generate 100 images and calculate the mean for each image. We order the means of the sampled images and reject the null hypothesis if the mean of the real data is too large or too small, compared to the sampled images, choosing as significance level  $\alpha=0.05$. For each data generator, we carry out 100 such tests.
> > > The proportion of rejected tests at significance level  $\alpha=0.05$ are:
> > >
> > >
> > > |    MNIST dataset | GAN\_MLP | DCGAN | VAE | NCSN | Real samples |
> > > | ---- |  ----------- | ----------- |----------- |----------- | ----------- |
> > > | mean statistic |       0.08| 0.03| 0.06| 0.05|  0.02|
> > >
> > >
> > > For all generating methods, the proportion of rejected tests is close to the significance level, although the different generators produce samples with considerable differences from the real data, which can easily be spotted visually. This example illustrates that, using the mean value in a Monte Carlo test instead of as a test statistic in NP-KSD is not powerful enough to distinguish the generators from the real sample. This finding is in contrast to Table 1 in the paper, which shows that NP-KSD and the mean-conditioned variant NP-KSD\_m reject almost all tests for the synthetic data generators. Hopefully, this answers your question; if we misunderstood your question, please let us know.

---

### Official Review · Reviewer_P9SF · 2022-07-12

**Rating:** 6
**Confidence:** 3
**Soundness:** 3 good
**Presentation:** 3 good
**Contribution:** 3 good

**Summary:**

The paper considers the problem of assessing implicit generative models using kernelised Stein discrepancy (KSD).
The paper introduces a non-parametric (NP) Stein operator to allow implicit models to be used in KSD (which wasn't possible due to the fact KSD requires at least the unnormalised density of the model).
Results on both synthetic and real datasets compared to MMD (which also only needs samples to compute) show that the proposed NP-KSD method has a better test power than MMD.

**Questions:**

- In algorithm 1, are we effectively learning the score function so to estimate KSD?
  - Is this what the author(s) mean(s) in Line 232-234 when criticising MMD?
- In algorithm 2, do we effectively use more samples from G compared to MMD?

**Limitations:**

Limitations and potential negative societal impact are discussed.

**Strengths And Weaknesses:**

Pros
- The paper proposes a novel method that extends KSD to implicit models.
- The paper is well written and technical. The paper is not hard to follow.
- The paper is a solid contribution to better access the generation quality of generative models. This is becoming more important as synthetic data generation is seen as an important tool to solve privacy.
- The results look promising. It would be interesting to include a neural sampler trained by MMD, which would be an even stronger indication of the stronger test power of NP-KSD.

Cons
- Discussion compared to MMD is not enough; see my Qs below.

---

> ### Author Response · Authors · 2022-07-31
> **Sample size and concerns about MMD**
>
>
> Thank you for the comments and suggestions. We are pleased that you appreciate the importance of the problem as well as our contribution to its solution.
>
> The main issue arising seems to be the discussion of comparison with MMD.
> Indeed, similarly to MMD, we also consider a two-sample problem. MMD can be very useful when two sets of samples are of comparable size.
> A key difference to MMD is that in our setting one of the sample sizes ($n$, the observed data) is usually quite small and cannot easily be increased. In an extreme case, only one sample (such as one image) may be available.
> In contrast, the size of the other sample size, $N$, can be chosen as large as desired, as this set of samples is generated by the synthetic data generator $G$. Thus, the test situation is very imbalanced in sample size. In Algorithm 2, we may use as many samples from $G$ as is desired; and our theoretical results Proposition 3.1 and Theorem 3.2 underpinning the procedure are valid in the regime $N\rightarrow \infty$ with $n$ fixed, and Theorem 3.2 gives a bound on the rate of convergence.
>
> This imbalance makes MMD not suitable for a comparison method.
> The text in lines 232-234 alludes to issues arising in MMD when the sample size is small.
> In particular, in Appendix F.1 we illustrate that the MMD is not consistent in this highly imbalanced situation; Table 5 shows that the type-1 error in an MMD test does not reach the correct level. The somewhat terse sentence in line 249 points to the reasons why MMMD is not compared against. We have now expanded this sentence, to read
> ``We note that  MMD tests do not have controlled type-I error when $n\ll N$, thus are not suitable in this setting. However, MMD based methods to compare and criticise two generative models have been explored [Lloyd and Gharahramani, 2015, Sutherland et al. 2017].
> Detailed discussions and illustrations why  MMD tests are
> not included in the comparison list are found in Appendix F.1.''
>
> Instead, we compare with MMDAgg, a non-asymptotic test which is consistent even when $n\ll N$, see for example Table 5 in Appendix F. For this comparison, Algorithm 2 uses the same number of samples from  $G$ as  MMDAgg, to ensure a fair comparison. In our experiments shown in Figure 1,  NP-KSD based tests (in red with dots in Figure 1) outperform MMDAgg (in orange with triangles in Figure 1) in terms of test power.
>
> Regarding your question whether in Algorithm 1, we effectively learn the score function so to estimate KSD, this is indeed a high-level summary. A key novelty is that we estimate the KSD for the generator $G$, which can be carried out to arbitrary precision as we can generate as many samples from $G$ as desired; Proposition 3.1 gives the theoretical justification. Naively estimating this score function would be very difficult in high dimensions. Instead, we estimate univariate conditional score functions and combine them in Equation (7) to yield a Stein operator, which is the basis of the KSD test. When it is possible to sample from the conditional distributions (which we do not assume to be the case), a similar idea has been carried out in Singhal, R., Han, X., Lahlou, S., and Ranganath, R. (2019). Kernelized complete conditional Stein discrepancy. arXiv preprint arXiv:1904.04478.
> Not citing this paper was an oversight which has now been amended.
>
> We hope that this response addresses your questions and alleviates your concerns.

---

> > ### Comment · Reviewer_P9SF · 2022-08-07
> > **Reply to author response**
> >
> > The clarification on the sample size and more elaborated discussion on MMD vs NP-KSD clears my original concerns.
> > I suggest you to blend your discussion here into the main paper, which would help the presentation.

---

> > > ### Author Response · Authors · 2022-08-08
> > > **Blended discussion in the main text**
> > >
> > > Many thanks for your suggestions. We have amended the text to further include the discussion in the revised version. In the main text,
> > > Line 249, we now explicitly point out why MMD is not compared against and include the pointer to the Appendix F.1 where detailed illustrations and experiments are given. The expanded text reads:
> > >
> > >
> > > ``We note that MMD tests do not have controlled type-I error when $n \ll N$, thus are not suitable in this setting. However, MMD-based methods to compare and criticise two generative models have been explored [Lloyd and Gharahramani, 2015, Sutherland et al. 2017]. Detailed discussions and illustrations of why MMD tests are not included in the comparison list are found in Appendix F.1.''
> > >
> > > Appendix F.1 also shows that MMD is not consistent in this highly imbalanced situation; Table 5 shows that its type-1 error does not reach the correct level. Instead, we compare to MMDAgg because it is a non-asymptotic test with controlled type-1 error even when $n \ll N$, see Table 5 in Appendix F. Fig.1b assesses the effect of the number $N$ of generated samples (on the x-axis). As discussed in lines 260-265, the power of NP-KSD increases faster with sample size than that of MMDAgg.

---

### Meta-Review · Area_Chair_tEho · 2022-08-20

**Recommendation:** Accept
**Confidence:** Certain

**Metareview:**

Decision: Accept

This paper introduces a non-parametric (NP) Stein operator to allow implicit models to be used in KSD. So this enable the use of KSD for evaluating the performance of implicit models, and the new test statistic shows better test power compared to MMD test.

Reviewers commended that the paper writing is clear, and the contribution is solid and novel. There were a few technical concerns regarding the proposed KSD as well as comparisons to MMD, which were mostly addressed in author-reviewer discussions.

In revision for camera ready, I'd encourage the authors to include the additional experiments & discussions provided in the author feedback. Perhaps adding more MMD-based test baselines would strengthen the paper even further.

**Award:**

No

---

### Decision · Program_Chairs · 2022-09-14

Accept